# Accurate Interpolation for Scattered Data through Hierarchical Residual Refinement

**Shizhe Ding**[1,2], **Boyang Xia**[1,2], **Dongbo Bu**[1,2,3*]

[1] Key Laboratory of Intelligent Information Processing,
Institute of Computing Technology, Chinese Academy of Sciences, Beijing, China
[2] University of Chinese Academy of Sciences, Beijing, China
[3] Central China Institute for Artificial Intelligence Technologies, Zhengzhou, China
`{dingshizhe19s,xiaboyang20s,dbu}@ict.ac.cn`

## Abstract

Accurate interpolation algorithms are highly desired in various theoretical and engineering scenarios. Unlike the traditional numerical algorithms that have exact zero-residual constraints on observed points, the neural network-based interpolation methods exhibit non-zero residuals at these points. These residuals, which provide observations of an underlying residual function, can guide predicting interpolation functions, but have not been exploited by the existing approaches. To fill this gap, we propose **H**ierarchical **INT**erpolation Network (**HINT**), which utilizes the residuals on observed points to guide target function estimation in a hierarchical fashion. HINT consists of several sequentially arranged lightweight interpolation blocks. The first interpolation block estimates the main component of the target function, while subsequent blocks predict the residual components using observed points residuals of the preceding blocks. The main component and residual components are accumulated to form the final interpolation results. Furthermore, under the assumption that finer residual prediction requires a more focused attention range on observed points, we utilize hierarchical local constraints in correlation modeling between observed and target points. Extensive experiments demonstrate that HINT outperforms existing interpolation algorithms significantly in terms of interpolation accuracy across a wide variety of datasets, which underscores its potential for practical scenarios.

## 1 Introduction

Scattered data interpolation aims to estimate the unknown values of a continuous function at some target points based on a finite set of observed data points [11]. This technique plays a significant role in extensive theoretical and engineering domains, *e.g.,* partial differential equations (PDEs) solving, physical analysis and thermal engineering, *etc* [7, 20, 28, 2]. For instance, in thermal engineering, scattered data interpolation is employed to estimate the environmental temperature distribution of buildings [28] and electronic devices [2] based on temperature sensor readings obtained at dispersed points. In many applications, the function distributions are complex and the scattered points are non-uniformly distributed, which poses significant challenges for accurate interpolation algorithms.

Traditional scattered data interpolation algorithms follow a well-established pipeline: firstly a suitable set of basis functions is prepared (*e.g.* polynomial, Spline and radial basis functions), then a possible target function is generated by constructing a linear combination of these basis functions [11]. Basis function are selected depending on application scenarios, which highly relies on domain knowledge and human experience. This makes traditional algorithms hardly able to generalize between problems

---

*Corresponding author.

37th Conference on Neural Information Processing Systems (NeurIPS 2023).

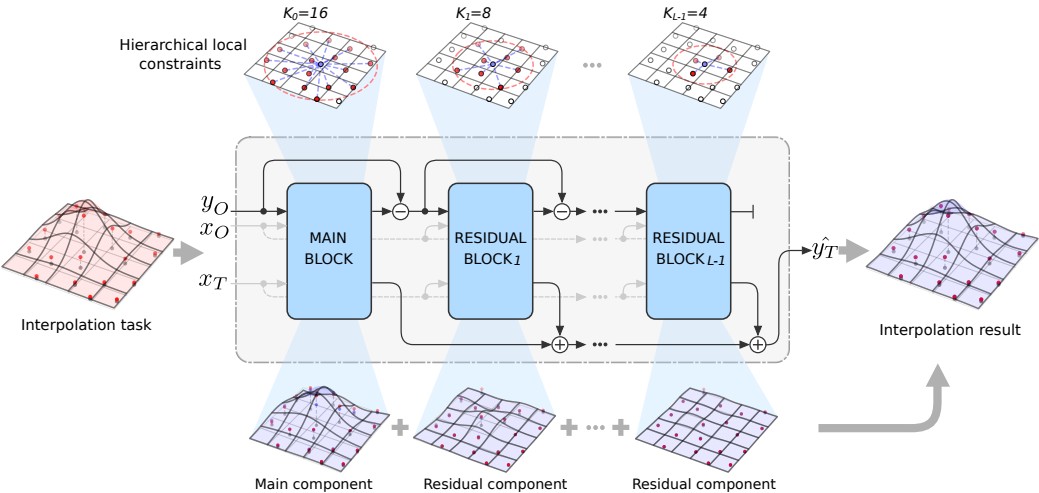

Figure 1: Overview of our HINT architecture. The architecture comprises multiple interpolation blocks, including a main block and several residue blocks. The main block directly estimates the primary components of the target function. Subsequent residue blocks refine the estimation based on the residuals on the observed points computed from previous blocks. As the blocks that output finer residuals place more emphasis on capturing fine-grained details of the function, we incorporate hierarchical local constraints on a valid range of observed points correlated with target points on these interpolation blocks.

and imposes much burden of searching and tuning. Thus, when targeting slightly more complex functions, these methods usually present limited interpolation accuracy.

Recently, the evolution of deep learning technology [17, 15, 16, 12, 3] has given rise to a plethora of neural network-based interpolation algorithms. Notable examples of such models encompass Neural Processes [8, 14, 19, 18], TFR-Transformer [2], and NIERT [4]. These models leverage neural network modules to capture the correlation between observed points and target points, and estimate the target point values based on the correlations and the observed point values. In contrast to traditional methods that rely on hand-crafted and rigid basis functions, these neural interpolators can adaptively discern the underlying distribution of the target function from a function set or scattered data set, consequently achieving superior generalization performance.

Different from the traditional interpolation algorithms that have exact zero-residual constraints on observed points (target function curve passes through the points) [11], the neural networks-based approaches exhibit non-zero residuals at these points. We assume these residuals represent a type of residual function, which maintains residual information sharing between observed points and target points. Although existing methods utilize the correlations between observed points and target points as conditions, few methods explore the use of residual information sharing between them.

To fill this gap, we introduce **H**ierarchical **INT**erpolation network (**HINT**) to utilize observed points residuals to guide target function estimation in a hierarchical fashion (depicted in Figure 1). Our key insight is that the target function estimation can be decomposed into a main function component and several residual function components. With this in mind, we design the framework of HINT as a hierarchy of lightweight interpolation blocks, among which the first block (main block) estimates the main component of the target function and subsequent blocks (residual blocks) predict the residual components in a progressively refining manner. In each residual block, to utilize residual information sharing between observed and target points, we predict the residuals of target points from accessible observed point residuals and via correlation modeling between two point domains. Finally, the main component and residual components of target points are accumulated to form interpolation results.

Furthermore, we consider that finer residual prediction for a target point requires more focused attention on its spatially neighboring observed points. Thus, we incorporate hierarchical local

constraints that progressively narrows down the valid range of observed points correlated with target points, to reduce the noise of correlation modelling during residual prediction.

We summarize our contributions as follows: 1) We propose a novel hierarchical residual refining framework, HINT, for scattered point interpolation. We introduce a series of residual blocks that utilize the residual information of observed points to guide target points prediction via correlation modeling in a coarse-to-fine fashion, to enhance interpolation accuracy. 2) We impose hierarchical local constraints on the range of observed points correlated with target points to conform to hierarchical residual scales, thereby further improving interpolation robustness. 3) We validate our framework on four scattered data interpolation task sets across diverse theoretical and application scenarios, which demonstrates our HINT surpasses existing state-of-the-art interpolation methods in terms of accuracy in various tasks consistently.

## 2 Method

### 2.1 Problem Statement and Notations

Given $n$ observed points with known values, denoted as $O = \{(x_i, y_i)\}_{i=1}^n$, and $m$ target points with values to be estimated, represented by $T = \{x_i\}_{i=n+1}^{n+m}$, the objective of the interpolation task is to accurately estimate the values $f(x)$ for each target point $x \in T$, where $f$ is the underlying function determined by the information from the observed points in $O$.

Here, the position of a point is indicated by $x_i \in \mathbb{R}^{d_x}$, while the value of a point is signified by $y_i = f(x_i) \in \mathbb{R}^{d_y}$. The function mapping is denoted by $f : \mathbb{R}^{d_x} \to \mathbb{R}^{d_y}$, with $d_x$ and $d_y$ representing the dimensions of a point's position and value respectively. The function $f$ originates from a function distribution, which can be implicitly represented by a set of scattered data.

For convenience, we denote the positions of the observed points as $x_O = \{x_i\}_{i=1}^n$, the values of the observed points as $y_O = \{y_i\}_{i=1}^n$, and the positions of the target points as $x_T = \{x_i\}_{i=n+1}^{n+m}$.

### 2.2 Hierarchical Residual Refining Framework

The architecture of our proposed HINT is illustrated in Figure 1. Our key insight is the target function estimation can be decomposed into a series of function components at different scales, including a main function component and several residual components. In light of this view, we design a hierarchical interpolation architecture, consisting of a series of $L$ neural network interpolation blocks. The first block (main block) serves as a coarse interpolator, estimating the main component of the target function, and subsequent $L - 1$ blocks (residual blocks) function as finer interpolators, estimating the residual parts of the target function.

The main block takes the positions and ground truth function values of the observed points, $x_O$ and $y_O$, along with the positions of the target points $x_T$, as inputs. It outputs the coarse function value predictions at both the observed and target points $\hat{y_O}^{(1)}, \hat{y_T}^{(1)}$.

$$\hat{y_O}^{(0)}, \hat{y_T}^{(0)} = \text{MainBlock}(x_O, y_O, x_T). \tag{1}$$

Then we can immediately obtain the estimated residuals of the target function at the observed points:

$$y_O^{(1)} = y_O - \hat{y_O}^{(0)}. \tag{2}$$

After that, the following $L - 1$ residual blocks predict the residual components of the target function in a progressively refining manner. Specifically, each residual block takes the residual of the previous block's predicted function values at the observed points as input and outputs the residual function values at both the observed and target points. The predicted residual function values at the observed points are used to compute the remaining residuals for the next residual block, while the predicted residual function values at the target points are employed to calculate the final function estimation values at the target points:

$$\hat{y_O}^{(l)}, \hat{y_T}^{(l)} = \text{ResidualBlock}^{(l)}(x_O, y_O^{(l)}, x_T), \tag{3}$$

$$y_O^{(l+1)} = y_O^{(l)} - \hat{y_O}^{(l)}. \tag{4}$$

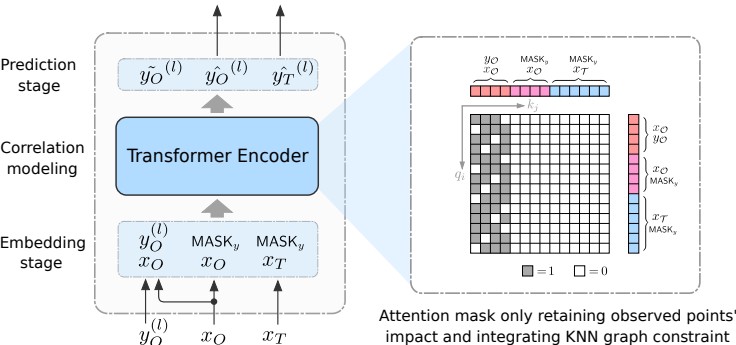

Figure 2: The proposed interpolation block structure. Given a specific interpolation task, each interpolation block starts by duplicating the observed points without values as *masked observed* points. Then, all scattered points are embedded. For the target points and masked observed points that are without values, a MASK token is used to fill the void. Next, a Transformer encoder is utilized to encode these scattered points in a unified fashion and model their correlations. Finally, the predictions are generated. To eliminate the target points' influence on the observed points and target points themselves, we apply a masked self-attention mechanism that exclusively preserves the impact of observed points. Meanwhile, to uphold the locality constraint of the interpolation task, we incorporate K-nearest neighbor graph (KNN graph) constraint in the attention mask.

Ultimately, we obtain the final interpolation function prediction by accumulating the initial main components and subsequent residual components:

$$\hat{y_T} = \sum_{l=0}^{L-1} \hat{y_T}^{(l)}. \tag{5}$$

## 2.3  Transformer-based Interpolation Block

Here we elaborate on the structure of our interpolation block illustrated in Figure 2, which is embodied as the main block and residual blocks of HINT. The $l$-th interpolation block takes the positions and values of the observed points $x_O$ and $y_O^{(l)}$, and the positions of the target points $x_T$ as inputs, and predicts the values of the target function at both the observed points and target points, i.e., $\hat{y_O}^{(l)}$ and $\hat{y_T}^{(l)}$. To facilitate the implementation of our hierarchical residual refining framework, we propose a novel encoder-only interpolation block building upon the basic structure of NIERT block [4], a state-of-the-art neural interpolation model. Please refer to Supplementary Material for a detailed description of NIERT. Specifically, we make two key modifications to the original NIERT block: **1)** A set of "masked observed" points are introduced to solve the problem of inconsistent distribution between observed points and target points. **2)** Local constraints are imposed on the range of observed points for target points in correlation modeling to accommodate different scales of the output function value. We introduce our interpolation block in detail as follows.

**Embedding Stage.** In the embedding phase, each point's position and value are separately embedded and concatenated to form the point's overall embedding. Following [4], we introduce a learnable token vector $\mathrm{MASK}_y$ to embed the absent value for target points. Although this avoids structural discrepancy between observed points and target points, there still exists the problem of inconsistent distribution between them. Moreover, observed point residuals and target point residuals serve as the input and output of the block, respectively. Thus consistent distribution between them helps the model to compute residuals more easily. To this end, we introduce *masked observed* points, which are the replicates of observed points without values. This simulates the absence of value in target points for observed points, ensuring the distribution of them is consistent with target points. After this replication operation. the embeddings of three sets of points can be represented as:

$$h_i^0 = \begin{cases} [\mathrm{Linear}_x(x_i), \mathrm{Linear}_y(y_i^{(l)})], & \text{if } i \in [1, n], & \triangleright \text{ observed points} \\ [\mathrm{Linear}_x(x_{i-n}), \mathrm{MASK}_y], & \text{if } i \in [n+1, 2n], & \triangleright \text{ masked observed points} \\ [\mathrm{Linear}_x(x_{i-n}), \mathrm{MASK}_y], & \text{if } i \in [2n+1, 2n+m] & \triangleright \text{ target points,} \end{cases} \tag{6}$$

where $i - n$ denotes index shifting.

**Correlation Modeling.** Then the embeddings of the points are fed into a stack of $P$ Transformer layers, to model the correlations between observed points and target points, and produce resultant points encodings. Through the $p$-th transformer layer, the encoding of the $i$-th point $h_i^p$ are transformed to $h_i^{p+1}$. Following NIERT [4], we use attention mask mechanism on vanilla self-attention in each Transformer layer, to avoid the unexpected interference of target points on observed points. Thus, all three set of points are treated as queries $\{q_i^p\}_{i=1}^{2n+m}$, while only observed points are treated as keys $\{k_i^p\}_{i=1}^n$ and values $\{v_i^p\}_{i=1}^n$, with these concepts referring to [27].

Then, the unnormalized attention score $e_{ij}^p$ can be computed via scaled dot product $e_{ij}^p = \frac{q_i^p \cdot k_j^p}{\sqrt{d_k}}$. Here, $d_k$ is the dimension of the query and key vectors. Based on prior knowledge that the observed points near the target points have a greater impact on the interpolation of target points than distant ones [7, 4], we impose controllable local constraints on the attention mask to accommodate different scales of residuals. We retain only the influence of the $K$ nearest observed points (under the Euclidean distance metric) of the $i$-th target point (we denote $\mathcal{N}_K(i)$ here), masking out the attention scores of distant observed points (see Figure 2):

$$\hat{e}_{ij}^p = \begin{cases} e_{ij}^p, & \text{if } j \in \mathcal{N}_K(i), \\ -\infty, & \text{otherwise.} \end{cases} \tag{7}$$

where $K$ is a hyperparameter for local constraints. Then the attention score can be obtained by softmax normalization $\alpha_{ij}^p = \frac{\exp(\hat{e}_{ij}^p)}{\sum_{j=1}^n \exp(\hat{e}_{i,j}^p)}$. With reasonable $K$, this design helps lower down noise in correlation modelling, thus improves interpolation accuracy.

**Estimating stage.** For each point $i \in [1, 2n + m]$, we estimate its value $\hat{z}_i$ through feeding its representation at the final Transformer layer into an MLP, $\hat{z}_i = \text{MLP}_{\text{out}}(h_i^P)$. These values are divided into three parts, including 1) predicted values of target function at target points $\hat{y_T}^{(l)} = \{\hat{z}_i\}_{i=2n+1}^{2n+m}$, 2) predicted values of target function at "masked observed" points $\hat{y_O}^{(l)} = \{\hat{z}_i\}_{i=n+1}^{2n}$, and 3) auxiliary reconstruction of observed points $\tilde{y_O}^{(l)} = \{\hat{z}_i\}_{i=1}^n$. We use $\hat{y_O}^{(l)}$ rather than $\tilde{y_O}^{(l)}$ to compute residuals $y_O^{(l)} - \hat{y_O}^{(l)}$ as the input of next residual block. This is because $\hat{y_O}^{(l)}$ have consistent distribution with $\hat{y_T}^{(l)}$, which helps the model conveniently utilize sharing residual information between two types of points in to predict $\hat{y_T}^{(l)}$. We use $\tilde{y_O}^{(l)}$ for auxiliary loss function optimization.

## 2.4 Hierarchical Local Constraints

We expand local inductive bias [7] of interpolation in a hierarchical way. We assume the main block focuses on relatively global information for it estimates the primary components of the target function, while residual blocks focus on local information for it estimates the residual components with smaller scales. Considering this, we impose hierarchical local constraints on the interpolation blocks by utilizing K-nearest neighbors constraints with decreasing $K$. Specifically, we establish a simple rule, setting two hyperparameters, $K^{(0)}$ and $K_{\min}$, where $K^{(0)}$ determines the $K$ value for the main block, and $K_{\min}$ represents the minimum $K$ value for the residual blocks. We recursively set the $K$ value for the $l$-th residual block as $K^{(l)} = \max\left(K^{(l-1)}/2, K_{\min}\right)$ (see Figure 1). It is worth noting that the specific values for $K^{(0)}$ and $K_{\min}$ depend on the characteristics of the dataset.

## 2.5 Loss Function

The loss function consists of two components: one for optimizing the interpolation error $\mathcal{L}_{\text{int}}$, and the other as an auxiliary term $\mathcal{L}_{\text{aux}}$, *i.e.*,

$$\mathcal{L} = \mathcal{L}_{\text{int}} + \lambda \mathcal{L}_{\text{aux}}, \tag{8}$$

where $\mathcal{L}_{\text{int}} = \text{Error}(\hat{y_T}, y_T)$ is the primary objective and $\mathcal{L}_{\text{aux}} = \frac{1}{L} \sum_{l=0}^{L-1} \text{Error}(\tilde{y_O}^{(l)}, y_O^{(l)})$ is the auxiliary reconstruction term. The primary objective is to minimize the discrepancy between the predicted function values and the ground truth at the target points. And the auxiliary term serves to constrain the auxiliary reconstruction at observed points for each interpolation module, facilitating

the accurate reconstruction of both observed and target points. Here, The error function used in this context can be either Mean Squared Error (MSE) or Mean Absolute Error (MAE), depending on the precision metric requirements of the specific scenario. $\lambda$ is the loss balance hyperparameter.

# 3   Experimental Setting

We evaluate the interpolation performance of HINT on four representative datasets and compare it with existing state-of-the-art approaches. We describe these datasets, the baselines used for comparison and the experimental setup as follows. Please refer to the Supplementary Material for more details.

## 3.1   Datasets

The four datasets are Mathit-2D and Perlin for synthetic function interpolation, TFRD for temperature field reconstruction, and PTV for particle tracking velocimetry. We provide a brief introduction to these four datasets, and further details and statistical information can be found in the Supp.

**Theoretical Dataset I: Mathit-2D**. Mathit-2D [4] contains interpolation tasks derived from two-dimensional synthetic symbolic functions. 512 data points are sampled within the square region $[-1, 1]^2$ and are randomly divided into observed and target points to form an interpolation task. The number of observed points is randomly chosen from the range [10, 50].

**Theoretical Dataset II: Perlin.** To evaluate the interpolation performance on functions with fine details, we additionally constructed the Perlin dataset, which is sampled from random two-dimensional Perlin noise functions [23]. Each interpolation task involves randomly sampling from a Perlin function defined on $[-1, 1]^2$, resulting in 64 observed points and 192 target points.

**Application Dataset I: TFRD.** TFRD [2] is a two-dimensional interpolation dataset for temperature field reconstruction. It contains three sub-datasets, ADlet, DSine, and HSink corresponding to different field boundary conditions. Each temperature field has 40,000 data points, with 37 points serving as observed points and the remaining points as target points to form interpolation tasks.

**Application Dataset II: PTV.** PTV [4] is a scattered dataset for real-world two-dimensional particle tracking velocimetry, where each instance corresponds to a set of particles moving in a velocity field extracted from the images of laminar jet experiment scenes [25]. Each instance contains 2,000 to 6,000 data points, from which we randomly select 512 points as observed points, while the remaining points serve as target points for prediction.

## 3.2   Baselines

The baselines for evaluation include: **1) Conditional Neural Processes (CNPs)** [8] utilize MLPs to predict function distributions given observed points. **2) Attentive Neural Processes (ANPs)** [14] leverage attention mechanisms to enhance CNPs' performance. **3) Bootstrapping Attentive Neural Processes (BANPs)** [19] which employ the bootstrap technique to further improve ANPs' performance. **4) TFR-Transformer** [2] incorporates typical encoder-decoder-style Transformers model correlations between observed points and target points. **5) NIERT** [4] leverages a Transformer encoder-only architecture to unify the processing of observed and target points in the same feature space to obtain better interpolation performance.

## 3.3   Experimental Setup

**Hyperparameter Setting.** We carefully select and tune several hyperparameters of HINT, including the number of interpolation blocks $L$, the number of attention layers in each block, the local parameter $K_0$, and the minimum local parameter $K_{\min}$. For the Mathit-2D dataset, we set $L = 2$, with 6 attention layers in the main block and 2 attention layers in the residue block. As for Perlin, PTV, and TFRD datasets, we set $L = 4$ with 2 attention layers in each block. The choice of $L$ was based on prior research and our experimental observations. To ensure a fair comparison, we maintained consistency between HINT and the baseline models, NIERT and TFR-Transformer, in terms of their key hyperparameter settings. Specifically, we keep the number of attention layers, the number of attention heads, and the dimensionality of the hidden space, consistent across all these models.

Table 1: Interpolation accuracy on Mathit dataset.

| Interpolation approach | MSE ($\times 10^{-4}$) on Mathit-2D test set |
|---|---|
| CNP | 24.868 |
| ANP | 14.001 |
| BANP | 8.419 |
| TFR-Transformer | 5.857 |
| NIERT | 3.167 |
| **HINT (ours)** | **2.903** |

Table 2: Interpolation accuracy on Perlin dataset.

| Interpolation approach | MSE ($\times 10^{-5}$) on Perlin test set |
|---|---|
| CNP | 48.642 |
| ANP | 23.731 |
| BANP | 20.737 |
| TFR-Transformer | 12.101 |
| NIERT | 7.185 |
| **HINT (ours)** | **5.848** |

Table 3: Interpolation accuracy on PTV dataset.

| Interpolation approach | MSE ($\times 10^{-3}$) on PTV test set |
|---|---|
| CNP | 137.573 |
| ANP | 32.111 |
| BANP | 33.585 |
| TFR-Transformer | 17.125 |
| NIERT | 5.167 |
| **HINT (ours)** | **3.507** |

Table 4: Interpolation accuracy on TFRD dataset.

| Interpolation approach | MAE ($\times 10^{-3}$) on TFRD test set | | |
|---|---|---|---|
| | HSink | ADlet | DSine |
| CNP | 204.351 | 91.782 | 92.456 |
| ANP | 164.491 | 54.684 | 58.589 |
| BANP | 59.728 | 28.671 | 19.107 |
| TFR-Transformer | 64.987 | 27.074 | 29.961 |
| NIERT | 23.519 | 3.473 | 8.785 |
| **HINT (ours)** | **13.758** | **1.761** | **4.912** |

Regarding the local parameter $K^{(0)}$, we set it to the number of observed points in the interpolation task for Mathit-2D, Perlin, and TFRD datasets, as these datasets have a relatively small number of observed points. For the PTV dataset, we set $K^{(0)}$ to 128 to accommodate its larger number of observed points. These values were selected to balance the local constraints with the available information in each dataset. Additionally, we set the minimum local parameter $K_{\min}$ to 8 for all datasets. This value was chosen to ensure a sufficient level of local modeling and capture fine-grained details in the interpolation process.

**Evaluation metric.** Following the precedent set by previous works [2, 4] and for the purpose of facilitating comparison, we adopted the MSE as the evaluation metric for Mathit-2D, Perlin, and PTV datasets. Additionally, for the TFRD dataset, we utilized the MAE as the evaluation metric.

## 4 Experimental results

### 4.1 Interpolation Accuracy

We present comprehensive comparisons of test set average interpolation accuracy of our interpolation approach, HINT, with state-of-the-art approaches on the four representative datasets. On the Mathit-2D dataset, we observe a notable enhancement of 8.34% in accuracy compared to the best method NIERT (Table 1). Similarly, on Perlin and PTV datasets, our method demonstrates remarkable accuracy gains of 30.23% and 18.61% respectively (Tables 2 and 3). Furthermore, on three subsets of the TFRD dataset, *i.e.*, HSink, ADlet and DSine, our method exhibits substantial accuracy gains of 41.50%, 49.29% and 44.08% (Table 4) over NIERT. These compelling results validate the efficacy of our approach on interpolation tasks across diverse datasets, highlighting the superiority and versatility of our algorithm in both theoretical and real-world application scenarios.

### 4.2 Qualitative Analysis

**Qualitative Comparison.** Here we provide a more intuitive comparison of the interpolation performance between HINT and existing interpolation methods. We randomly select an interpolation task from the Perlin test datasets, visualizing the interpolation results and error maps of HINT and existing interpolation methods. As shown in Figure 3, for this interpolation task from the Perlin test set, our proposed HINT achieves the highest interpolation accuracy (MSE $\times 1.04 \times 10^{-4}$). The error map of BANP exhibits strong error hotspots. TFR-Transformer and NIERT still have relatively large accuracy difference regions, while HINT has the largest area with minimal discrepancies (dark blue regions) on its error map.

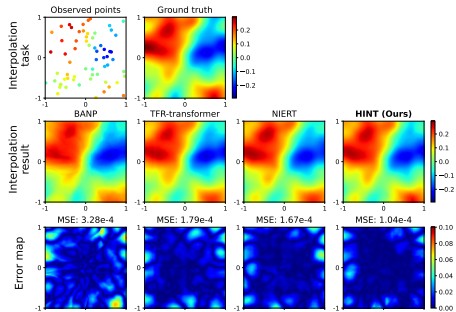

Figure 3: Qualitative comparison on a 2D interpolation task extracted from Perlin test set. The top row shows the 64 observed points and the ground-truth function. The second row shows interpolation functions reported by HINT and the state-of-the-art approaches. The last row shows their error maps (their differences with the ground truth).

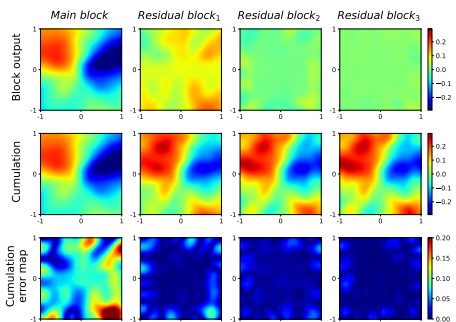

Figure 4: Visualization of output from each interpolation block on an example task (same as in Figure 3) from the Perlin test set. The top row of figures shows the output for each block, the middle row shows the accumulated output of these blocks, and the bottom row displays the error maps, indicating the differences between the cumulative predictions and the ground truth.

**Residual Visualization.** To investigate the role of each block in our proposed HINT, we visualized the output components of each block, as shown in Figure 4. We can find that the main block produces a relatively smooth overall function shape for the target function, lacking local details. The first residual block outputs more localized fluctuations in the target function with a smaller scale. The second residual block generates even smaller-scale components that appear to capture more local details and the last block contributes minuscule components. Observing the cumulative sum of the outputs from these blocks, we find that the accumulated function becomes progressively more detailed and closely resembles the ground truth of the target function as the outputs of the subsequent blocks are added. By examining the error maps of each cumulative sum, we observe that the output of the main block has strong error hotspots and subsequent blocks gradually refine the prior predictions until the final error map displays uniformly low errors throughout.

### 4.3 Ablation Study

**Ablation on Main Design Choices of HINT.** We validate the efficacy of the auxiliary loss term $\mathcal{L}$aux, masked observed points, and local constraints in HINT, as shown in the upper section of Table 5. When the auxiliary loss term is omitted (equivalent to setting the weight $\lambda$ of the auxiliary loss term $\mathcal{L}$aux to 0), as indicated in the table as "HINT w/o $\mathcal{L}_{\text{aux}}$", HINT experiences a reduction in accuracy on all three datasets, with a notably significant impact on Perlin and TFRD-ADlet (decreasing by 30% and 125%, respectively). Omitting the local constraints ("HINT w/o KNN") also leads to decreased accuracy on these datasets, particularly affecting Perlin and PTV (with a decrease of 20% and 43%, respectively). Additionally, we assess the effectiveness of masked observed points ("HINT w/o masked obs."). In this setup, instead of using $\hat{y_O}^{(l)}$, we utilize $\tilde{y_O}^{(l)}$ to compute residuals on the observed points. Experimental results in this configuration reveal a decline in accuracy across all three datasets, most notably on Perlin (a reduction of 55%), underscoring the significance of maintaining a consistent distribution between observed and target points. These findings highlight the advantages of incorporating these design choices within our framework.

The lower half of Table 5 presents the ablation results on different local constraints. For a straightforward evaluation, we employed two ablation settings where the local constraint parameter $K$ was set to either $K^{(0)}$ or $K^{(1)}$ for all interpolation blocks. Here, $K^{(0)}$ represents the local parameter of the first block in the hierarchical local constraint, while $K^{(1)}$ represents the local parameter of the second block. These two ablation configurations of local parameters represent a non-hierarchical approach with similar hyperparameters to our hierarchical approach. These results demonstrate the hierarchical design of local constraints has significant superiority over the non-hierarchical one in our method.

**Ablation on Interpolation Blocks Number.** Table 6 presents the results obtained by varying the number of blocks from 1 to 5 on two datasets, Perlin and PTV. The results indicate that when there

Table 5: Ablations on main design choices.

| Interpolation approach | Interpolation accuracy on | | |
|---|---|---|---|
| | Perlin (MSE$\times 10^{-5}$) | PTV (MSE$\times 10^{-3}$) | TFRD-ADlet (MAE$\times 10^{-3}$) |
| HINT w/o $\mathcal{L}_{\mathrm{aux}}$ | 7.630 | 3.656 | 3.985 |
| HINT w/o KNN | 7.017 | 5.001 | 1.944 |
| HINT w/o masked obs. | 9.071 | 3.547 | 2.773 |
| HINT | **5.848** | **3.507** | **1.761** |
| HINT ($K = K^{(0)}$) | 7.017 | 3.811 | 1.944 |
| HINT ($K = K^{(1)}$) | 8.083 | 4.791 | 2.975 |
| HINT (Hierarchical) | **5.848** | **3.507** | **1.761** |

Table 6: Ablation on block number $L$.

| Block number | Interpolation accuracy on | |
|---|---|---|
| | Perlin (MSE$\times 10^{-5}$) | PTV (MSE$\times 10^{-3}$) |
| 1 | 21.453 | 14.362 |
| 2 | 11.279 | 5.527 |
| 3 | 8.073 | 4.386 |
| 4 | **5.848** | **3.507** |
| 5 | 6.224 | 3.618 |

is only one block (i.e., no residual block), HINT performs poorly on both datasets. However, with the introduction of a single residual block (i.e., two blocks in total), the interpolation performance of HINT improves significantly. This highlights the effectiveness of incorporating the interpolation module. As the number of blocks increases, the interpolation accuracy of HINT shows incremental improvements with smaller margins. When the number of blocks reaches 5, the interpolation accuracy of HINT shows a slight decline on the Perlin dataset and remains relatively stable on the PTV dataset. These results suggest that after a certain number of blocks, the additional blocks no longer contribute to the learning of HINT, and the interpolation accuracy plateaus.

# 5 Related Work

**Traditional Approaches for Scattered Data Interpolation.** Traditional interpolation algorithms for scattered data construct the target function through linear combinations of explicit basis functions. For one-dimensional scattered data, techniques such as linear interpolation, Lagrange interpolation, Newton interpolation [11], and spline interpolation [9] are commonly employed. For more general scattered data residing in higher-dimensional spaces, Shepard's methods [26], and Radial Basis Function (RBF) interpolation [24, 6] are widely used. MIR [29] employs the Taylor series for interpolation and determines the weights by optimizing the interpolation error. In engineering applications, hierarchical interpolation methods have also been developed [5], which are often combined with spatial partitioning techniques or the selection of nearby points to reduce computational complexity. All these traditional methods are inherently non-adaptive, as they don't learn the function distribution from data. This restricts their interpolation precision and hampers their generalization across diverse scenarios.

**Learning-based Approaches for Scattered Data Interpolation.** Recent neural network-based scattered data interpolation algorithms can be broadly categorized into two groups: probabilistic and deterministic. Probabilistic methods, such as the family of Neural Processes [8, 14, 18, 30, 19, 21], estimate the conditional distribution of the target function based on observed points. Deterministic methods, such as TFR-Transformer [2] and NIERT [4], estimate the target function based on observed points with a focus on interpolation accuracy. Technically, these approaches use neural network modules to model the correlation between observed and target points, subsequently obtaining the representation and estimation of target point values. For instance, CNPs [8] employs MLPs to model the correlation between observed and target points, while ANPs [14], TFR-Transformer, and NIERT utilize attention mechanisms and Transformer layers [27] for the same purpose. Among these methods, NIERT employs the architecture of the Transformer encoder to uniformly handle observed and target points, resulting in enhanced interpolation accuracy. It stands as the current state-of-the-art neural interpolation algorithm, serving as inspiration for our interpolation block design. Notably, these current neural interpolators have thus far neglected the information regarding interpolation residuals at observed points, which constitutes a shared limitation.

**Residual Learning and Decomposition Structure.** Residual connections in neural networks can be traced back to residual modules in ResNet [10] and DenseNet [13]. In this module, the output of each stack is fed into subsequent stacks, for better optimization of deep neural networks. For a different propose, in time series forecasting, N-BEATS [22] presents a doubly residual framework with two distinct residual branches: one branch for backcast prediction and another branch for the forecast prediction at each layer. The residual between the backcast prediction and history input is

exploited to forecast prediction. N-HiTS [1] integrates hierarchical sparse prediction into the similar doubly residual framework, to enhance both forecasting accuracy and interpretability. However, these techniques are primarily tailored for fundamental time series signal decomposition and display a notable limitation when tasked with learning sparse and scattered data interpolation. Although the exploitation of residual is successfully applied in time series forecasting, it is hard to apply in scattered data interpolation straight-forwardly for sparsity and irregularity of scattered data.

# 6    Conclusion

In this paper, we propose HINT, a novel hierarchical residual refining framework for scattered data interpolation. By employing a series of residual blocks in a coarse-to-fine fashion and incorporating hierarchical local constraints in residual correlation modeling, HINT effectively improves interpolation accuracy. Our extensive validation across diverse theoretical and application scenarios demonstrates the superior performance of HINT compared to existing state-of-the-art interpolation methods. We acknowledge the limitations of our approach regarding the hyperparameter $K$ in hierarchical local constraints, and we plan to explore methods for learning adaptive $K$ values in future work.

# 7    Acknowledgements

We would like to thank the National Key Research and Development Program of China (2020YFA0907000), and the National Natural Science Foundation of China (32370657, 32271297, 82130055, 62072435) for providing financial supports for this study and publication charges. The numerical calculations in this study were supported by ICT Computer-X center.

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
