# Supplementary Material for Accurate Interpolation for Scattered Data through Hierarchical Residual Refinement

**Shizhe Ding**[1,2], **Boyang Xia**[1,2], **Dongbo Bu**[1,2,3*]

[1] Key Laboratory of Intelligent Information Processing,
Institute of Computing Technology, Chinese Academy of Sciences, Beijing, China
[2] University of Chinese Academy of Sciences, Beijing, China
[3] Central China Institute for Artificial Intelligence Technologies, Zhengzhou, China
{dingshizhe19s,xiaboyang20s,dbu}@ict.ac.cn

## A  Source Code

We have made the code for our implementation publicly accessible for the sake of transparency and reproducibility. It can be found at: `https://github.com/DingShizhe/HINT`.

## B  Additional Details of NIERT block and HINT block

In the present study, we introduce an enhanced interpolation block, predicated on the NIERT interpolator [2], and elucidate the principal improvements. To underscore the distinctions between the two methodologies, we furnish an in-depth exposition of the architectural design of the NIERT interpolator.

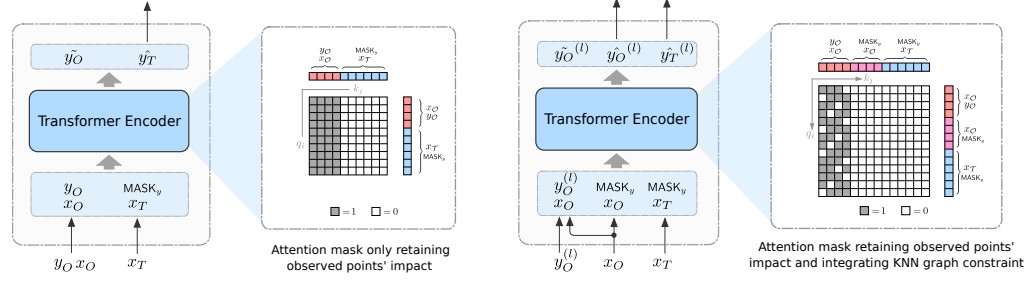

(a) NIERT interpolator.      (b) HINT interpolation block.

Figure 1: Comparison of architecture of NIERT interpolator and HINT interpolation block.

Fig. 1a depicts the NIERT interpolator, which accepts the positions and values $(x_O, y_O)$ of observed points and the position $x_T$ of the target point as inputs, and outputs the predicted value at the target points $\hat{y_T}$, along with reconstructed values at auxiliary observed points $\tilde{y_O}$.

In the embedding phase, NIERT uniformly embeds both observed and target points. A learnable mask vector is introduced for target points lacking value data. The NIERT interpolator's core is a Transformer encoder with a masked self-attention mechanism, uniformly encoding observed and target points and modeling their correlations within the same feature space. The masked self-attention

---

*Corresponding author.

37th Conference on Neural Information Processing Systems (NeurIPS 2023).

mechanism ensures that the target point's influence is excluded when modeling scattered correlations, aligning with the interpolation task's requirement that the target interpolation function is determined solely by the observed points. The Transformer encoder yields representations of encoded observed and target points containing value information. The NIERT interpolator's final output includes the predicted value at the target point $\hat{y_T}$, serving as the interpolation result.

The NIERT, a Transformer encoder-only architecture that uniformly encodes observed points and models their correlations, exhibits superior interpolation accuracy. Leveraging insights from the NIERT interpolator, we have devised the architecture of the interpolation block in HINT, as depicted in Fig. 1b. Our proposed architecture, specifically adapted to HINT's overall framework, introduces significant enhancements, underlining the following key aspects:

1. HINT employs residuals on observed points to estimate residuals on target points. To acquire residuals on observed points, the interpolation block must estimate consistent values on these points while also estimating values on target points. Consequently, each interpolation block introduces a set of 'masked observed' points to address the issue of inconsistent distribution between observed and target points.

2. HINT capitalizes on the hierarchical structure of the function's main and residual components and applies progressively stronger local constraints on these interpolation blocks. As such, each interpolation block requires the ability to configure local constraints. These constraints, implemented via K-nearest neighbor graph constraints, are applied to the range of observed points for target points in correlation modeling.

## C   Additional Datasets Details

### C.1   Dataset Statistics

Table 1 provides the statistical details for interpolation tasks in each dataset, encompassing the dimensionality of the scattered data, the count of observed points, and the count of target points. These sets of interpolation tasks, with their wide-ranging characteristics, facilitate a thorough assessment of interpolation performance.

Table 1: Statistics of the interpolation tasks used for training in each dataset.

| Dataset | $d_x$ | $d_y$ | #All points $N$ | #Observed points $n$ | #Target points $m$ |
|---------|-------|-------|-----------------|----------------------|--------------------|
| Mathit-2D | 2 | 1 | 256 | $[10, 50]^*$ | $N - n$ |
| Perlin | 2 | 1 | 256 | 64 | $N - n$ |
| PTV | 2 | 2 | $[2291, 5899]$ | 512 | $N - n$ |
| TFRD | 2 | 1 | 40000 | 37 | $N - n$ |

### C.2   Dataset Details

**Theoretical dataset I: Mathit-2D** [2] constitutes an interpolation task set designed for 2D mathematical function interpolation. This dataset was derived from random sampling of 1,000,000 uniquely generated two-dimensional mathematical functions. The domain of these functions is confined to $[-1, 1]^2$, with the range set to $[0, 1]$. For each task sampling, 512 scattered points were randomly chosen and partitioned into observed and target points. The count of observed points was randomly sampled within the range of $[10, 50]$. In addition to the training dataset, a test set comprising $12,000$ interpolation tasks was also synthesized. To prevent overlap between the training and test sets, mathematical expressions common to both sets were excluded from the test set. The synthesis code for Mathit-2D can be found at the following URL: `https://github.com/DingShizhe/NIERT`.

**Theoretical dataset II: Perlin** is another synthetic assembly of interpolation tasks, specifically designed for the numerical interpolation of two-dimensional rough functions. Each interpolation task in this dataset stems from randomly generated Perlin noise functions [5]. These functions are

---

*This implies that in the interpolation tasks of this dataset, the number of observed points $n$ varies from 10 to 50.

created by amalgamating multiple smooth and continuous noise functions of varying levels and frequencies, yielding rich details that present a formidable challenge for scattered point interpolation tasks. We synthesized 1024 random Perlin functions for training and an additional 1024 for testing. The domain of these functions is confined to $[-1,1]^2$. For each task sampling, 256 scattered points were randomly chosen and partitioned into 64 observed points and 192 target points. The synthesis code for Perlin can be accessed from our code repository at the following URL: `https://github.com/DingShizhe/HINT`.

**Application dataset I: TFRD** [1] is employed to evaluate the performance of 2D temperature field reconstruction from measurements taken at a finite set of scattered points. It comprises three distinct sub-datasets: HSink, ADlet, and DSine, each of which simulates the temperature field of mechanical devices using specific types of heat generation and boundary conditions [1]. Each reconstruction task within each subset encompasses $200 \times 200$ regular grid points, representing the temperature field within a $0.1m \times 0.1m$ square area. Of these points, the temperature at 37 scattered points is known and these are utilized as observed points, while the remaining points serve as target points. Each subset contains a total of $10,000$ training instances and $10,000$ test instances. The TFRD datasets can be downloaded from the following URL: `https://github.com/shendu-sw/recon-data-generator/`.

**Application dataset II: PTV** [2] is a scattered dataset where the scattered points represent particles with velocities in the flow field. PTV is utilized to assess performance in particle tracking velocimetry, i.e., the reconstruction of a two-dimensional velocity field from a finite number of observed particles with velocities. These data are extracted from raw images of laminar jet experiment scenes [7]. There are a total of 1200 raw frames, each corresponding to a distinct velocity field, and each frame extracts a set of scattered points with velocity, ranging from approximately 6000 points at most to 2000 points at least. For each set of scattered data, we randomly select $512$ points as observed points and the remaining points as target points to construct an interpolation task. We randomly allocate a quarter of the tasks to the test set and the remainder to the training set. The PTV dataset can be downloaded from the following URL: `https://github.com/DingShizhe/PTV-Dataset/`.

## D  Additional Implementation Details

**Hyperparameter Settings.** As delineated in Table 2, we outline the key hyperparameter configurations for our HINT methodology across all datasets. In an effort to ensure a fair comparison in each experiment, we have aligned the number of attention layers in the NIERT and TFR-Transformer methods with the aggregate number of attention layers across all interpolation blocks of HINT. Furthermore, we have upheld consistency in the number of heads in the multi-head attention mechanism and the dimensions of the latent space.

Table 2: Hyper-parameters of HINT in experiments on the four datasets.

| Parameter name | Mathit-2D | Perlin | PTV | TFRD | | |
| --- | --- | --- | --- | --- | --- | --- |
| | | | | HSink | ADlet | DSine |
| Number of interpolation blocks $L$ | 2 | 4 | 4 | 4 | 4 | 4 |
| Number of layer of main block | 6 | 2 | 2 | 2 | 2 | 2 |
| Number of layer of residual block | 2 | 2 | 2 | 2 | 2 | 2 |
| Number of heads | 8 | 4 | 4 | 4 | 4 | 4 |
| Hidden dimension | 512 | 128 | 128 | 128 | 128 | 128 |
| Local constraint parameter $K^{(0)}$ | $n^*$ | $n$ | $n/4$ | $n/2$ | $n$ | $n$ |
| Local constraint parameter $K_{\min}$ | 8 | 8 | $n/8$ | 8 | 8 | 8 |
| Auxiliary loss weight $\lambda$ | 0.05 | 0.01 | 0.0001 | 0.5 | 0.5 | 0.5 |

In our experiments, we discerned that the calibration of hyperparameters profoundly influences the interpolation efficacy of the model. Venturing into novel interpolation contexts, the paramount hyperparameter determinations in our model can be construed based on the ensuing insights:

1. **Block number** ($L$): Functions manifesting smoother profiles are optimally served with a diminutive $L$. In juxtaposition, functions characterized by meticulous details or higher frequency components demand a magnified $L$.

2. **Attention layer number in the main block**: Voluminous datasets call for an escalated number of attention layers to ensure optimal model capacity. Conversely, for datasets of a more diminutive scale, curtailing attention layers can mitigate overfitting risks.

3. **Local constraint parameter** ($K_0$, $K_{\min}$): The calibration of $K$ is non-trivial. For functions displaying expansive spatial auto-correlation, augmented $K_0$ and $K_{\min}$ are propitious. In our experiments, we found that determining the optimal $K_0$ and $K_{\min}$ often requires trial and tuning.

For instance, for the Mathit-2D dataset—a synthetic set comprised of smoother mathematical functions with a significant volume—a smaller block number $L$ and more attention layers seem optimal. The mathematical functions' features correlate over longer distances (periodicity, symmetry, trends, etc.), thus initially a larger $L$ covering all observation points is prudent. Conversely, for the Perlin dataset—a real-world 2D velocity field dataset of smaller volume but intricate, detailed functions—a larger block number $L$, fewer attention layers, and a smaller $L$ are logical.

**Training Parameter Settings.** We optimize model parameters utilizing the Adam optimizer [3]. For the Mathit-2D dataset, our training protocol involves 160 epochs with a fixed learning rate of 0.0001 and a batch size of 256. For the Perlin dataset, we adopt a similar approach, employing a batch size of 128 and a training period of 100 epochs, also with a non-decaying learning rate of 0.0001. For the PTV dataset, we modify our strategy to accommodate a batch size of 4, a learning rate of 0.0005 with a decay rate of 0.97 per epoch, and a training period of 100 epochs. Finally, for the three sub-datasets of TFRD, we implement a batch size of 5, a learning rate of 0.0005 with a decay rate of 0.97, and a training duration of 100 epochs.

Our models were implemented using the PyTorch framework [4]. All experimental procedures were conducted on a computational infrastructure equipped with two NVIDIA GeForce RTX 3090 GPUs.

# E Computational Efficiency Comparison

## E.1 Comparison of Theoretical Computational Efficiency

Table 3 presents a comparison of the theoretical computational efficiency of our proposed HINT with the comparable method NIERT. The average theoretical floating-point operations (GFLOPs) on the entire dataset are used as the metric. It can be observed that on the PTV dataset, HINT achieves a lower average theoretical floating-point operation count of 3.968 compared to the state-of-the-art interpolation method NIERT (5.421). On the Perlin dataset, HINT has a higher average theoretical floating-point operation count of (0.238) compared to NIERT (0.178). On the TFRD-ADlet dataset, HINT has a higher average theoretical floating-point operation count of (27.083) compared to NIERT (24.009).

These results indicate that HINT offers competitive computational efficiency in terms of theoretical floating-point operations compared to existing high-precision methods. On the PTV dataset, HINT demonstrates superior computational efficiency compared to NIERT. On the Perlin and TFRD-ADlet datasets, HINT's computational efficiency is slightly lower than NIERT, but within an acceptable range and in the same order of magnitude.

Table 3: Comparison of computational efficiency (GFLOPs) on different datasets.

| Interpolation approach | Computational efficiency on dataset (GFLOPs) | | |
|---|---|---|---|
| | PTV | Perlin | TFRD-ADlet |
| NIERT | 5.421 | 0.178 | 24.009 |
| HINT (ours) | 3.968 | 0.238 | 27.083 |

---

[*]Here, $n$ is the number of observed points of the interpolation task.

It's imperative to highlight that even though the adoption of local constraints, in theory, reduces computational demands, HINT consistently manifests a higher FLOP count than NIERT across specific datasets, an observation clearly corroborated by the comparison in Table 3. While at first glance this disparity might be perplexing, it is cogently rationalized by HINT's intricate architectural choices. On one hand, HINT's structure is characterized by multiple blocks, each distinctively furnished with embedding and prediction layers, setting it in stark contrast to NIERT's more linear configuration. On the other hand, HINT encompasses an extensive token repertoire, weaving together both observed and masked data points, as well as target entities, aggregating to $(2n + m)$ tokens. In contrast, NIERT adeptly navigates a more concise token collection, tallying $(n + m)$, where $n$ and $m$ explicitly denote the quantity of observed and target points, respectively.

## E.2 Comparison of Empirical Interpolation Times

We conducted additional evaluations to assess the efficiency of our proposed method, HINT. Specifically, we evaluated the average interpolation time required by our HINT method in comparison to existing interpolation techniques, encompassing both traditional and deep learning-based approaches, on the Mathit-2D test dataset. The inclusion of traditional methods was motivated by the well-known efficiency of conventional algorithms in interpolation tasks, serving as a reference benchmark. In accordance with the methodology outlined in [2], we selected Radial Basis Function (RBF) interpolation [6] and MIR [8] as representatives of traditional interpolation algorithms. The results, comprising the average interpolation time and interpolation accuracy on the test dataset, are presented in Table 4.

Table 4: The average time and accuracy comparison of interpolation approaches on Mathit-2D dataset.

| Interpolation approach | Average interpolation time (ms) | Interpolation accuracy (MSE$\times10^{-4}$) |
|---|---|---|
| RBF | 0.66 | 34.706 |
| MIR | 80.30 | 27.460 |
| CNP | 13.53 | 24.868 |
| ANP | 27.55 | 14.001 |
| BANP | 29.19 | 8.419 |
| TFR-Transformer | 26.16 | 5.857 |
| NIERT | 31.99 | 3.167 |
| HINT (ours) | 36.20 | 2.722 |

As observed in the table, the traditional interpolation method, RBF, exhibits the highest interpolation efficiency at 0.66 ms, albeit with a noticeable compromise in accuracy. On the other hand, MIR offers slightly improved accuracy but has the longest interpolation time among all evaluated methods, standing at 80.30 ms. In the realm of deep neural network-based techniques, a discernible trend emerges, indicating a progression in accuracy from CNP to our HINT method, accompanied by a gradual increase in computational time. Specifically, our HINT method boasts an average time of 36.20 ms, marginally higher than that of NIERT (31.99 ms), yet substantially more efficient than the traditional MIR (80.30 ms).

Furthermore, it is noteworthy that our current HINT implementation employs dense attention computation combined with an attention mask to implement the hierarchical local constraints outlined in our paper. Given that these local constraints are theoretically sparse, there remains room for optimizing the computational cost of HINT. Our primary research focus was centered on achieving superior interpolation accuracy, and at this stage, HINT emerges as the optimal choice for scenarios that prioritize high accuracy with some flexibility regarding time constraints.

## F  Additional Evaluation on High-dimensional Scattered Data

In light of the critical importance of validating the accuracy of interpolation algorithms in high-dimensional data, we introduced a 10-dimensional dataset referred to as "D10" to assess the interpolation performance of our proposed HINT method. Within this dataset, each function results from the summation of multiple 10-dimensional Gaussian functions, formalized as:

$$f(x) = \sum_{k=1}^{K} A_k \exp\left(-\frac{1}{2}\frac{(x-c_k)^2}{\sigma_k^2}\right).$$

We maintained a fixed $K$ value of 5. The parameters for each Gaussian function were uniformly sampled: the center $c_k$ from $[-1,1]^{10}$, the width $\sigma_k$ from $[1,2]$, and the weight $A_k$ from $[-1,1]$.

Our training set comprised 256K instances, while the test set contained 512 cases, all sampled from this function distribution. Each instance consisted of 64 observed points and 192 target points, uniformly sampled from $[-1,1]^{10}$. For the test set, we conducted direct evaluations of classical methods and data-driven models after 100 training epochs. The interpolation accuracy of these methods is presented in Table 5.

Table 5: Interpolation performance on D10 dataset.

| Interpolation approach | Interpolation accuracy (MSE$\times 10^{-4}$) |
|---|---|
| CNP | 35.623 |
| ANP | 12.578 |
| BANP | 12.077 |
| TFR-Transformer | 7.465 |
| NIERT | 5.496 |
| HINT (ours) | 4.173 |

These results suggests that HINT consistently outperforms other methods on the D10 dataset, achieving the lowest MSE, which underscores HINT's capability to maintain high precision even in high-dimensional scenarios.

## G   Additional Examples of Interpolation Results

To provide a more comprehensive demonstration of the interpolation performance of our proposed HINT, we present the interpolation results on additional test set examples, contrasting these with the results from existing methodologies, as shown in Fig. 2 3 4 and 5. The cases we highlight below are drawn from the test sets of Mathit-2D, Perlin, PTV, and TFRD-ADlet. In the visualizations for each example, the image in the top-left corner displays the ground truth for the case, while the image in the bottom-left corner illustrates the observed points for the interpolation task. The row of images on the top right presents the interpolation results from HINT and existing methods, and the row of images on the bottom right provides the error maps comparing these interpolation results with the ground truth.

In the majority of examples, the interpolation accuracy of HINT significantly outperforms that of the compared baselines. Visually, the error map of HINT's interpolation results is superior to the error maps of other methods' interpolation results, demonstrating its exceptional interpolation performance.

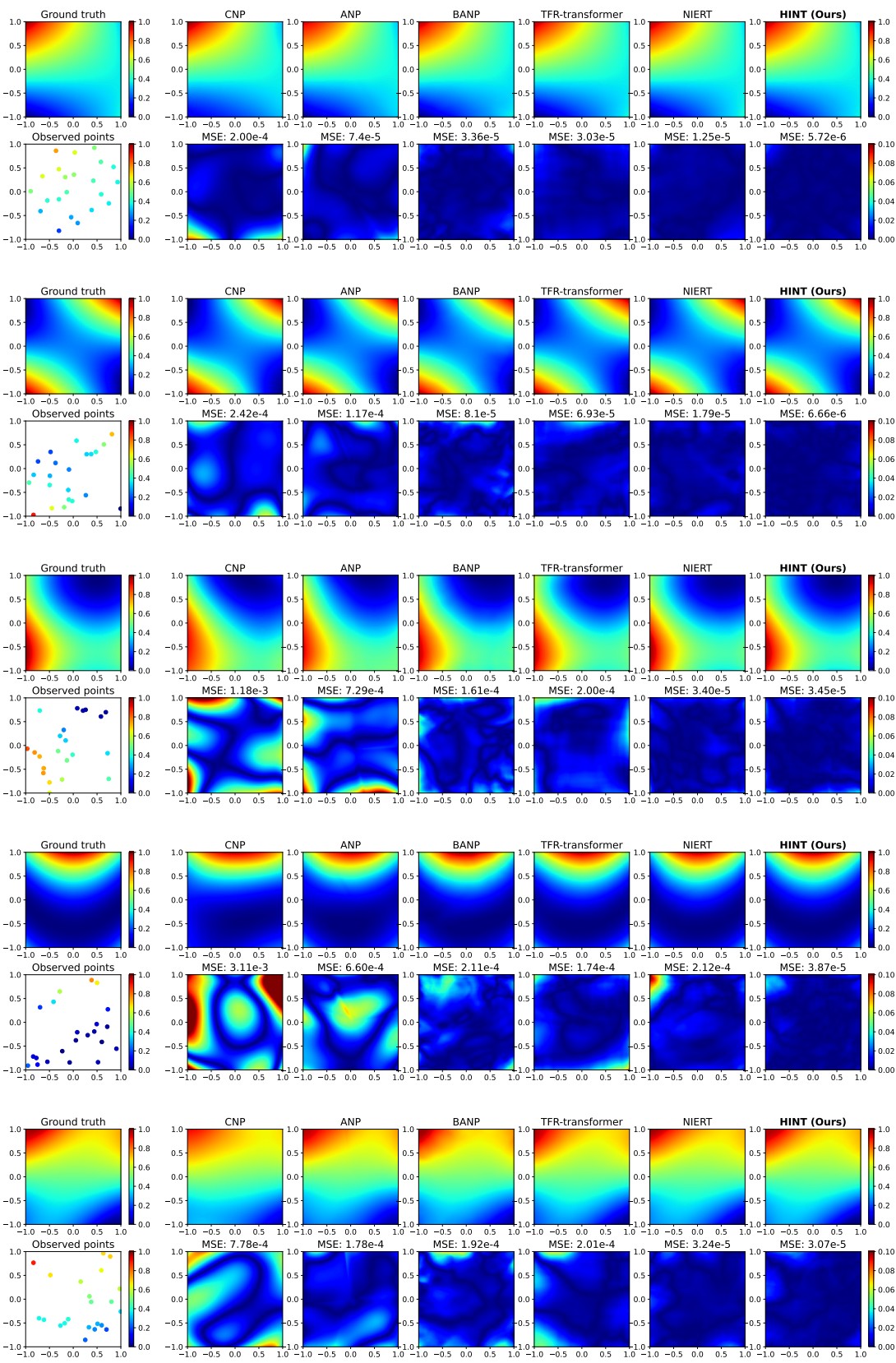

Figure 2: Additional cases from Mathit-2D test set.

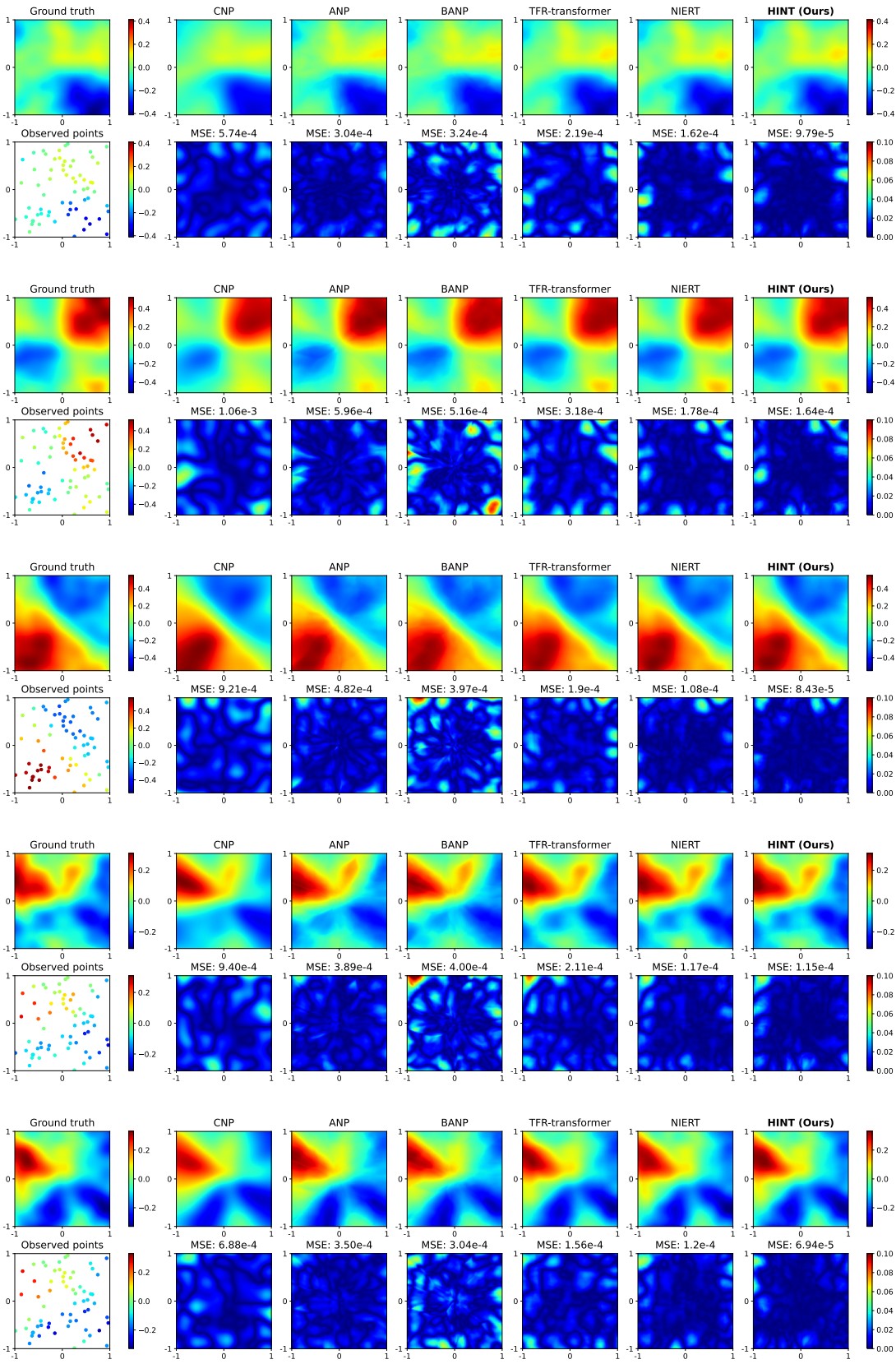

Figure 3: Additional cases from Perlin test set.

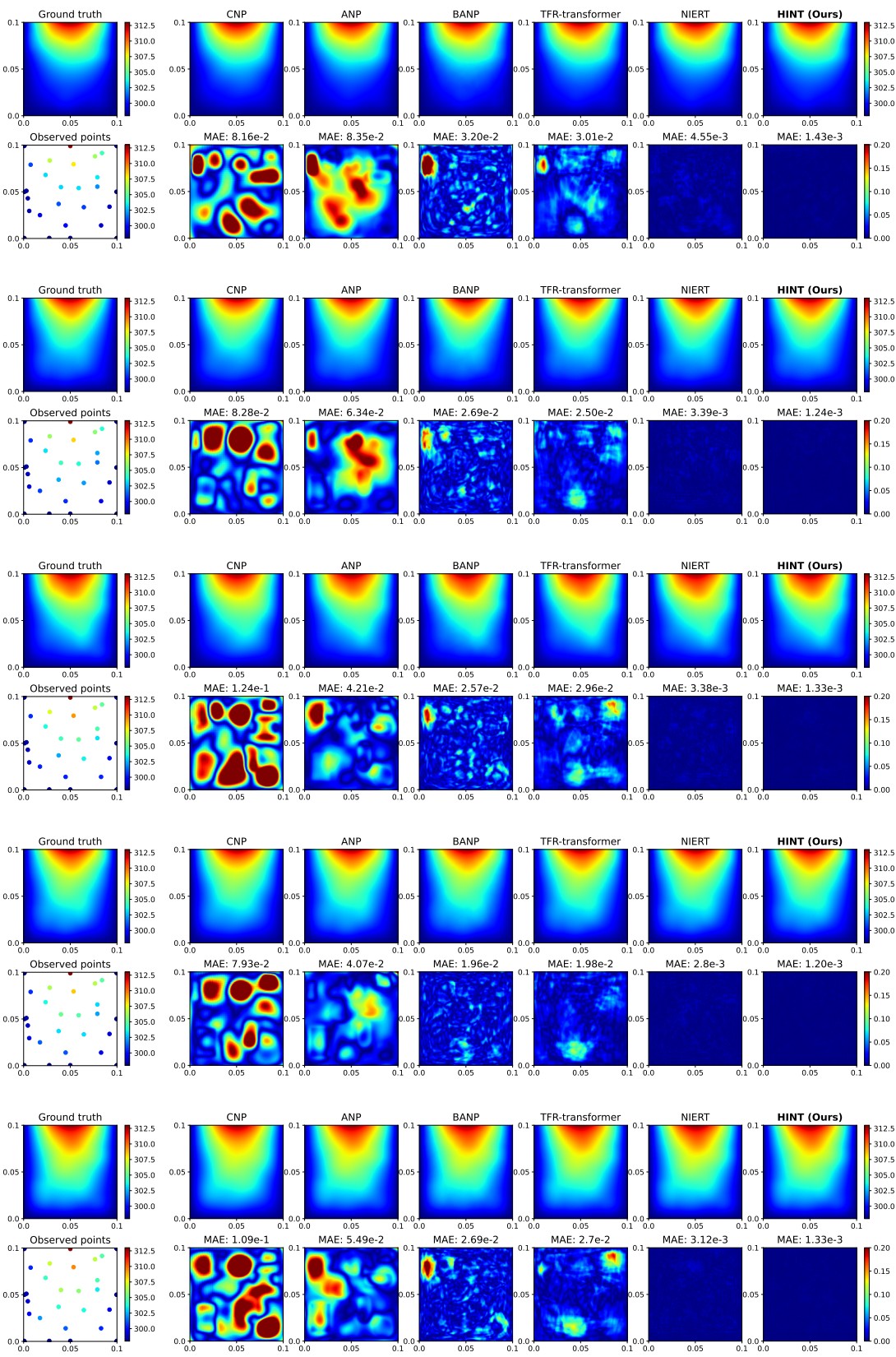

Figure 4: Additional cases from TFRD-ADlet test set.

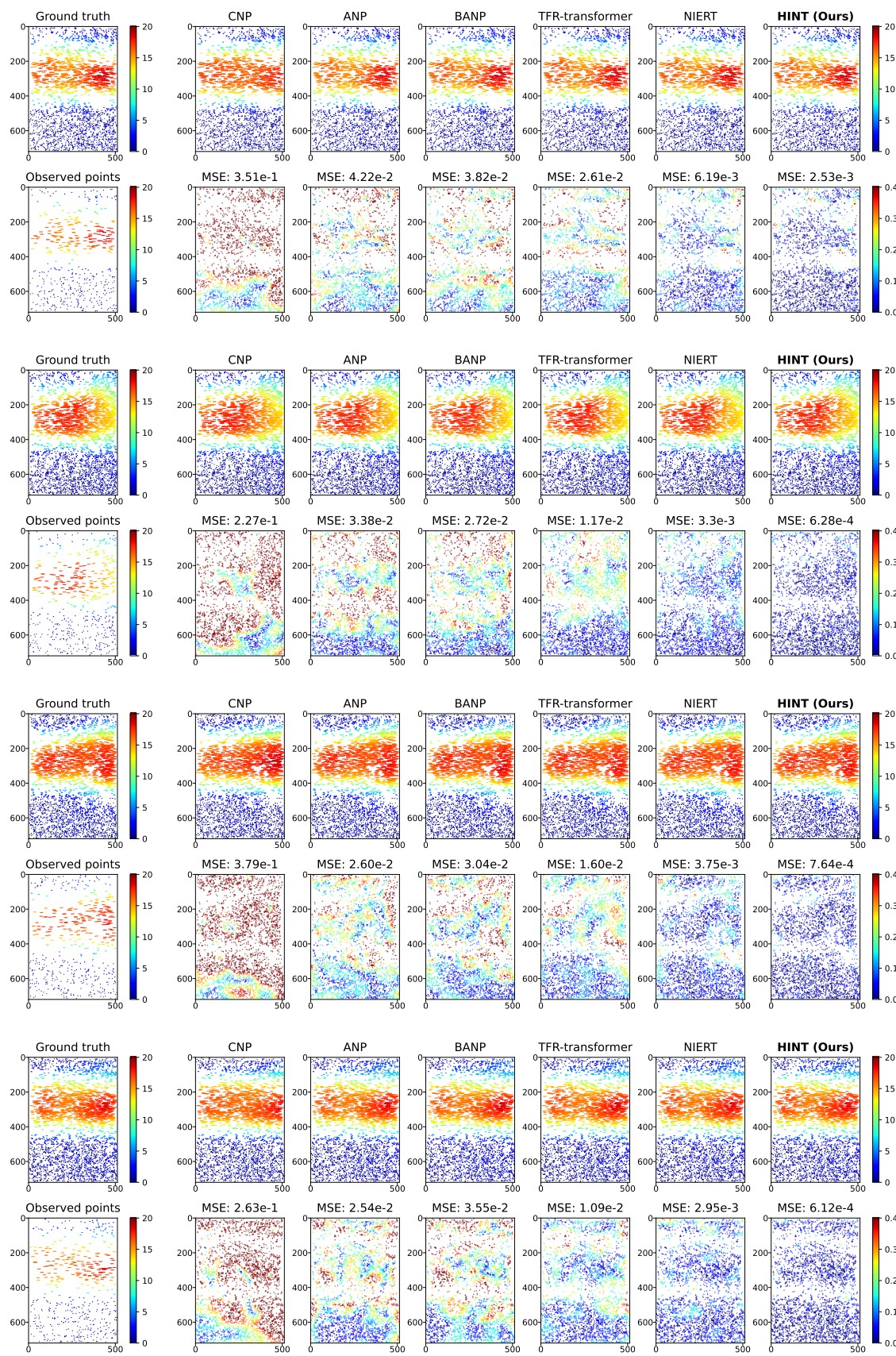

Figure 5: Additional cases from PTV test set.