# OpenReview forum: "Accurate Interpolation for Scattered Data through Hierarchical Residual Refinement"
_NeurIPS.cc/2023/Conference — NeurIPS 2023 poster_

### Official Review · Reviewer_WnAW · 2023-07-05

**Soundness:** 3 good
**Presentation:** 2 fair
**Contribution:** 2 fair
**Rating:** 3
**Confidence:** 4

**Summary:**

This paper introduces the Hierarchical INTerpolation Network (HINT), a hierarchical framework that leverages the residuals on observed points to guide the estimation of the target function. HINT comprises multiple lightweight interpolation blocks arranged sequentially. The first block estimates the main component of the target function, while subsequent blocks predict the residual components using the residuals from the preceding blocks. By accumulating the main component and residual components, HINT produces the final interpolation results. Experiments demonstrate the effectiveness of HINT, and the results demonstrate HINT can get better performance compared with existing interpolation algorithms across a wide range of datasets.

**Strengths:**

1) This paper is easy to follow.
2) The Ablation Study clearly validates the effectiveness of the proposed auxiliary loss term, masked observed points and local constraint in HINT.

**Weaknesses:**

1) The main concern is the novelty of this paper is not clearly presented. The major contribution of the paper seems to be the proposed hierarchical residual refining framework but its novelty has not been sufficiently addressed. It is worth noting that previous studies, specifically [1] and [22] as mentioned in the last paragraph of the section of Related Work, have already proposed hierarchical residual architectures for time series forecasting. However, the paper does not discuss the challenges in applying the hierarchical residual refining framework to the field of interpolation, regarding that interpolation can also be viewed as prediction from existing observations. It neither emphasizes the distinction between the proposed architecture and existing architectures. As a result, the relation between this paper and existing literature is not clear and the main contribution of the paper cannot be evaluated.
2) The presentation is not clear and should be carefully revised. For example, the positions of the observed points are defined twice in Line 83 and Line 84. The expression “It outputs and outputs the coarse function value predictions at both the observed and target points” in Line 94 is wrong.
3) The experimental results provided in the paper are limited. For instance, the authors did not conduct experiments to investigate the performance of the proposed method with varying ratios of observed points. Therefore, it is hard to evaluate the effectiveness of the proposed method under different data conditions.


**Questions:**

The motivation of the proposed hierarchical residual refining framework is not clear, could you explain the motivation of hierarchical residual refining framework from the perspective of experiments, such as ablation study, or theoretical analysis?

**Limitations:**

The limitations of this paper regarding the hyperparameter K in hierarchical local constraints.

---

> ### Author Rebuttal · Authors · 2023-08-09
>
> ### Response to Reviewer WnAW
>
> **Comment 1:** The paper's novelty, especially regarding its hierarchical residual refining framework, isn't clearly articulated, with insufficient distinction from previous works on hierarchical residual architectures.
>
> **Response 1:**  Thank you for your thoughtful comments and concerns regarding the novelty of our work. We'd like to address these concerns systematically:
>
> 1. **Main contributions of our HINT**:
>     1. Our paper's key contribution is a novel hierarchical residual refining framework for scattered data interpolation. While inspired by N-Beats [22] and N-Hits [1], our model diverges significantly from both them in its application to this specific task (as detailed below).
>     2. The hierarchical local constrained approach progressively refines residual predictions, leading to reduced residuals and improved interpolation accuracy.
>     3. Our model has outperformed existing methodologies across all datasets tested.
>
> 2. **Differences between scattered data interpolation and time series forecasting**:
>     1. Data Dimensionality: Time series forecasting is 1D, focusing on modeling long-term dependencies. Scattered data spans 2D or higher, revealing complex spatial relationships.
>     2. Sparsity: Time series data are densely recorded within time frames, whereas scattered data is often sparse.
>     3. Ordering: Time series data is consistently ordered over intervals, whereas scattered data is randomly spaced, complicating holistic approaches.
>     4. Function Priors: Time series often exhibit cyclical and trending patterns, as underscored in [1] and [22]. Scattered data rarely has such clear priors, and when it does, its random distribution complicates application.
>
> 3. **Differences between our HINT and N-Beats [22] and N-Hits [1]**:
>     1. On methodology: Both our method and [1] & [22] use a dual residual structure for signal decomposition. However, while [1] & [22] categorize time series signals into components like cyclical and trending, facilitating prediction, our HINT refines by capturing diminishing residual functions, aligning closer with function approximation.
>     2. Basic block implementation: The predictive blocks in [1] & [22], tailored for time series, harness its orderly and dense nature, using MLPs to predict signal base weights. Such networks are not ideal for scattered data interpolation. In contrast, our approach uses a modified Transformer encoder, leveraging self-attention to model correlation between observation and target points.
>     3. On hierarchical approach: [1] decomposes by sampling and interpolating the whole time series signals based on various cycle scales. Our hierarchy, rather than down-sampling scattered data, introduces local constraints during correlation modeling, diverging from [1]'s methods.
>
> We hope this clarifies the distinctions between our work and prior art, emphasizing the novelty of our approach. We appreciate the opportunity to elaborate on our contributions and look forward to any further queries or comments you might have.
>
> **Comment 2:** Typos at line 83 and 94.
>
> **Response 2:** Thanks for pointing out the typos.
>
> 1. The end of line 83 will be corrected from "the positions of the observed points" to "the values of the observed points".
>
> 2. We'll rectify the redundancy in Line 94.
>
> Apologies for any oversight. We'll revisit the manuscript for thoroughness and clarity.
>
>
> **Comment 3:** Lack of experiments to investigate the performance of the proposed method with varying ratios of observed points.
>
> **Response 3:** Thank you for raising this valid concern.
>
> We concur that evaluating the performance of our method with varying ratios of observed points would enrich our validation process. We have conducted supplementary experiments on the Mathit-2D dataset to gauge the impact of different numbers of observation points on interpolation accuracy, as depicted in the figure in [global response](https://openreview.net/forum?id=8d9wVXri89&noteId=RMsTw2cPZh).
>
> These findings further underscore the superiority of our approach. Not only does it excel in terms of average interpolation accuracy, but it also outperforms existing interpolation algorithms across all levels of observed point numbers. These results will be appended to the supplementary materials for further clarity.
>
> **Comment 4:** Motivation of the proposed hierarchical residual refining framework is not clear, explain the motivation of hierarchical residual refining framework from the perspective of experiments
>
> **Response 4:** Thank you for inquiring about the motivation of our hierarchical residual refining framework.
>
> 1. Experimentally, Sec.4.2's Residual Visualization (on Pages 7 and 8) offers insight, especially Fig. 4. It depicts the outputs of each block, their cumulative effects, and the error map on a Perlin test set interpolation task.
>
>     1. The Main block primarily captures the interpolation task's essence but misses finer details. Subsequent Residual Blocks progressively capture increasing detail levels, with the third block being especially subtle.
>
>     2. As we layer the residual predictions, the cumulative output more closely aligns with the ground truth (refer to Fig. 3's left column), diminishing the error map.
>
> 2. Theoretically, scattered data interpolation often involves intricate details. Our framework, based on residual prediction, segments the interpolation task, easing the challenge. This tiered approach better captures the target function, outperforming previous techniques by fully utilizing information of observed point values, enhancing accuracy.
>
> ---
>
> [1] Cristian Challu, et al. N-hits: Neural Hierarchical Interpolation for Time Series Forecasting. arXiv preprint arXiv:2201.12886, 2022
>
> [22] Boris N Oreshkin, et al. N-beats: Neural basis expansion analysis for interpretable time series forecasting. arXiv preprint arXiv:1905.10437, 2019.

---

> > ### Comment · Reviewer_WnAW · 2023-08-13
> >
> > Thanks to the authors for their efforts in preparing the rebuttal. The authors have partially addressed my concerns. In particular, the performance of the proposed method with varying ratios of observed points is validated.  However, the novelty of this paper is still not clearly addressed or demonstrated. Therefore, I am to keep my rating.
> >
> > The authors contribute the main difference between this paper and related existing works to applications into different data structures. This paper is claimed to address the problem of scattered data interpolation, while related works like [1] and [22] focus on time series forecasting. The difference between these two data structures is then elaborated. However, there still lacks an in-depth analysis or demonstration on such difference, either in theory or with empirical study. For example, how does the sparse data structure motivate the proposed method and what kind of different designs is required for sparse data structure compared to dense data structure or data without a fixed form (e.g., images)? Although capturing diminishing residual functions and transformer encoder are claimed to specifically solve the randomness of the scattered data but the relationship is not clearly explained and validated. Furthermore, this paper is of limited technical novelty, especially on the design of network architectures. The core difference between the proposed modified Transformer encoder for basic block implementation and existing methods such as TFR-Transformer [2], NIERT [4], and ANPs [14] is not clarified. In the response to Reviewer ce6Q, the authors claimed that NIERT [4] and HINT exhibit significant differences in architecture and methodology. However, using a series of lightweight interconnected blocks to replace the singular Transformer encoder in NIERT (architecture), and adopting a hierarchical architecture to progressively improve the results (methodology), are common techniques in deep learning. I cannot identify significant differences compared to existing methods, especially for the top conference like NeurIPS.

---

> > > ### Author Response · Authors · 2023-08-21
> > > **Response to Reviewer WnAW**
> > >
> > > We appreciate the reviewer's efforts and comments. While we respect their perspective, we respectfully disagree on certain aspects. We'd like to address three specific concerns: the connection between our method and the scattered data structure, the novelty of our method, and the effectiveness of our approach.
> > >
> > > 1. **Connection to Scattered Data Structure:** Works on time series like [1] and [22] are tailored for ordered, dense, and contiguous time series data. In stark contrast, our method is specifically designed for scattered data interpolation, inherently and perfectly accommodating its structure. The arbitrariness and sparsity of scattered data necessitate its encoding and processing as a set, which is why we adopted the Transformer-based block, leveraging the permutation invariance property of its attention mechanism. We believe this is a widely recognized principle in scattered data interpolation research. Furthermore, tasks related to pixel interpolation, such as super-resolution and image completion, are vastly different in nature from scattered data interpolation. The substantial differences in data structure and distribution between them are evident and, we believe, require no exhaustive discussion.
> > >
> > > 2. **Novelty of Our Approach:** The reviewer perceives similarities between our basic block and NIERT and feels our overall framework echoes that of [1] and [22] in time series forecasting. Such an assessment, we argue, is an oversimplification. Our significant enhancements to the basic interpolation block, tailored for the residual prediction framework, shouldn't be overlooked, as detailed in Sec. 2.3 and the Supplementary material. This clearly differentiates it from NIERT. Moreover, our hierarchical residual refinement framework, combined with the *hierarchical local constraints*, is an integral and effective advancement that should be acknowledged. Additionally, within the domain of scattered data interpolation, we are the first to introduce such a hierarchical residual refinement framework, further underscoring our method's novelty.
> > >
> > > 3. **Effectiveness of Our Approach:** Our method has outperformed existing approaches in interpolation accuracy across multiple datasets, advancing the realm of high-precision scattered data interpolation. Moreover, the output objective components of each block transition in intensity from global to local scales, aligning with our hypothesis and exhibiting partial interpretability. This accomplishment warrants recognition.
> > >
> > > In summary, we're grateful for the reviewer's diligence and contribution. We hope our clarifications provide a more lucid and balanced evaluation of our work.
> > >
> > > -----
> > >
> > > [1] Cristian Challu, et al. N-hits: Neural Hierarchical Interpolation for Time Series Forecasting. arXiv preprint arXiv:2201.12886, 2022
> > >
> > > [22] Boris N Oreshkin, et al. N-beats: Neural basis expansion analysis for interpretable time series forecasting. arXiv preprint arXiv:1905.10437, 2019.

---

### Official Review · Reviewer_ce6Q · 2023-07-05

**Soundness:** 3 good
**Presentation:** 4 excellent
**Contribution:** 2 fair
**Rating:** 5
**Confidence:** 3

**Summary:**

The authors proposed an algorithm named Hierarchical Interpolation Network (HINT) to predict unseen point values in scattered data to replace the usage of manually designed interpolation algorithms. The HINT used the residuals on observed points to guide target function estimation and the hierarchical local constraints in correlation modeling between observed and target points. Experiments show that it outperforms existing interpolation algorithms in accuracy on 3 synthetic datasets.

-- Post rebuttal:
I read the rebuttal from the auhtors. It addresses some of my concerns on the comparsions and I would like to keep my score.

**Strengths:**

The algorithm leverages a neural network to adaptively discern the underlying distribution of the target function from a function set or scattered dataset, achieving state-of-the-art performance on four datasets.

**Weaknesses:**

1. Limited novelty in network design. Mask mechanism and partial attention are used in NIERT. The Embedding Stage and Correlation Modeling in Transformer blocks are very similar to NIERT.
2. Inadequate method comparisons. The paper compared Neural Processes algorithms proposed in 2018, 2019, and 2020 but the later developments in 2022 not included such as [1] and [2].
3. It did not use PhysioNet dataset which is compared in NIERT.

In SM:
1. Did not explain the reason why it has more FLOPs than NIERT in some datasets.
2. Mistake： In SM line 105, ‘On the PTV dataset,’ should be ‘On the Perlin dataset,’

[1] Wang, Qi, Marco Federici, and Herke van Hoof. "Bridge the Inference Gaps of Neural Processes via Expectation Maximization." The Eleventh International Conference on Learning Representations. 2022.
[2] Liu, Huafeng, et al. "Learning Intrinsic and Extrinsic Intentions for Cold-start Recommendation with Neural Stochastic Processes." Proceedings of the 30th ACM International Conference on Multimedia. 2022.


**Questions:**

How many samples should be observed at least to guarantee the accuracy of your algorithm and this interpolation task? Is there any theory supported by its lower limit?

Can it be used for more dimensional or more complex interpolation？ Such as interpolation for pixels (inpainting, super-resolution, video interpolation)?


**Limitations:**

The algorithm and the task setting did not consider the situation of incorrect observation (noisy) points which is common in theoretical and engineering domains.

---

> ### Author Rebuttal · Authors · 2023-08-09
>
> ### Response to Reviewer ce6Q
>
> We are grateful to the reviewer for the comprehensive comments. The following responses are structured to address each comment.
>
> **Comment 1:** Limited novelty in network design..
>
> **Response 1:** Our HINT and NIERT exhibit significant differences:
>
> 1. **On architecture**: NIERT is built upon a singular Transformer encoder as its core model. HINT employs a sequence of lightweight interpolation blocks based on the Transformer encoder, which are interconnected through residual connections.
>
> 2. **On methodology**: NIERT's approach involves directly predicting the function's values at target points. HINT strategy employs a hierarchical refinement process. It sequentially predicts the primary components of the function and a series of residual components and combines them.
>
> 3. **Adaptations from NIERT**: Our interpolation block is inspired by NIERT. Yet, we've made substantial adjustments to fit our broader architecture:
>    - Masked observed points as input is introduced, enabling a re-prediction of the function values at the observed points. This, in turn, provides us with a function's residual at observed points, which guides the subsequent residual function predictions.
>    - Local constraints applied in correlation modeling, which ensures that our interpolation block can interpolate the function's details at different scales.
>
> We believe our enhancements distinctly set HINT apart from NIERT, adding unique value and novelty. We trust this clarifies the concerns raised.
>
> **Comment 2:** Inadequate method comparisons..
>
> **Response 2:** 1. After thorough research on recent NPs algorithms, we observed that most either mirror methods we've compared or aren't apt for accuracy comparisons, as outlined below.
>
> 1. **Regarding [1]**: We previously examined this method. After a detailed review on its appendix and [review feedback](https://openreview.net/forum?id=A7v2DqLjZdq&noteId=19Gw_u0rbIy), we observed its accuracy weren't notably superior to techniques like CNP. Since we've already benchmarked against CNP, ANP, and BANP, we felt it redundant to include this method in our comparisons.
>
> 2. **Regarding [2]**: We noted that this method applies NPs algorithms to recommendation systems. Consequently, it was not considered relevant for comparison in the context of our study due to its weak correlation.
>
> 3. **Highlight on Notable Recent Work**: It's worth noting that among the recent developments, Transformer Neural Processes (TNP) [3] stands out as a significant contribution. However, its architecture is almost identical to NIERT. To avoid redundancy, we chose not to include it in our comparisons.
>
> In summary, we've selected methods for comparison that are leading figures within NPs algorithms. Combined with NIERT and TFR-Transformer, we believe they offer a balanced comparison.
>
> **Comment 3:** It did not use PhysioNet dataset.
>
> **Response 3:** We omitted the PhysioNet dataset primarily because it's a 1D time series. As noted in [4], scattered data interpolation typically involves data points irregularly spaced in 2D or higher spaces. While NIERT assessed their model on PhysioNet, given our research focus, we didn't find it essential for our evaluation.
>
> **Comment 4:** why more FLOPs than NIERT in some datasets?
>
> **Response 4:** The two primary reasons are:
> 1. HINT encompasses several blocks, each embedded with an embedding and a prediction layer. NIERT only has a embedding and a prediction layer.
> 2. The input token consideration for HINT is greater, encompassing tokens from both the observed and the masked observed points in addition to the target points, leading to (2n+m) tokens, while NIERT uses (n+m) tokens.
>
> **Comment 5:** Typos.
>
> **Response 5:** The error has been corrected as suggested.
>
> **Comment 6:** On the number of observed points at least to guarantee the accuracy. Is there any theory supported by its lower limit?
>
> **Response 6:** We observed that a decrease in the number of observed points correlates with an increase in interpolation error. Thus, to maintain our model's accuracy, the minimum required observed points would align with those in our test sets.
>
> Regarding a theoretical foundation, crafting such a framework is complex and outside our study's current scope. Real-world data often eludes neat theoretical frameworks. Hence, pinpointing a direct theoretical link between observed point counts and our method's accuracy remains challenging. We appreciate your emphasis on the theoretical aspect, which undeniably stands as a crucial avenue for future research.
>
> **Comment 7:** On high-dimensional interpolation or more complex interpolation for pixels.
>
> **Response 7:** Regarding your question on high-dimensional interpolation, Reviewer 746Q posed a similar suggestion. Please refer to our response to 746Q where we've included pertinent experiments.
>
> Pixel-related interpolation tasks quite differs from scattered data interpolation. These tasks usually involve data points uniformly distributed on a dense grid, imbued with rich semantic information. Adapting our method, which is tailored for scattered data interpolation, to these tasks would likely necessitate substantial alterations.
>
> **Comment 8: ** Limitation in handling noisy data
>
> **Response 8:** Our study centers on high-precision interpolation, assuming observation points are accurate. Addressing noisy data poses unique challenges and falls outside this research's purview. We recognize its significance and might explore it in subsequent works.
>
> ---
>
> [1] Wang, Qi, et al. Bridge the Inference Gaps of NPs via Expectation Maximization. ICLR. 2022.
>
> [2] Liu, Huafeng, et al. Learning Intrinsic and Extrinsic Intentions for Cold-start Recommendation with Neural Stochastic Processes. ICM. 2022.
>
> [3] Nguyen, Tung, et al. Transformer Neural Processes: Uncertainty-aware meta learning via sequence modeling. arXiv, 2022.
>
> [4] Wendland, Holger. Scattered data approximation. Vol. 17. Cambridge university press, 2004.

---

> > ### Author Response · Authors · 2023-08-21
> > **Response to Reviewer ce6Q**
> >
> > We sincerely thank the reviewer and the AC for their efforts. We hope our response adequately addresses and clarifies the concerns raised.

---

> ### Comment · Area_Chair_HHeT · 2023-08-19
>
> Dear Reviewer,
>
> Could you please acknowledge that you have reviewed the author's response? If you have decided to maintain or revise your score, kindly provide a brief explanation if the rebuttal has or hasn't addressed your initial concerns adequately.
>
> Your prompt feedback is essential to the evaluation process.
>
> Best regards, Your AC

---

### Official Review · Reviewer_RpoB · 2023-07-06

**Soundness:** 3 good
**Presentation:** 2 fair
**Contribution:** 3 good
**Rating:** 5
**Confidence:** 3

**Summary:**

This work presents a new transformer based approach for scattered data interpolation. By extending NIERT [4] in several aspects, authors achieved improved results in the target task. The extensions include 1) hierarchical residual refinement to produce fine-grained interpolation results and 2) hierarchical local constraints to constrain the set of observed points used in the interpolation block. In short, unlike existing approaches, authors attempt to conduct the scattered data interpolation in a progressive fashion by estimating interpolation residuals within gradually narrowing search space.

**Strengths:**

1) The proposed components were reasonably designed to obtain fine-grained interpolation results.
2) Intensive ablation studies support the effectiveness of these components.


**Weaknesses:**

1) The datasets used in experiments consists of two-dimensional data. Performance should be evaluated with sparse dataset of much higher dimension.
2) Some explanations were given in the embedding stage, but it is very difficult to interpret why the masked tokens for the given observed points are necessary.
3) The local constraint K may be problematic, as it can be dependent on the data properties of each application.


**Questions:**

Please consider the weaknesses 1), 2), 3).

**Limitations:**

N.A.

---

> ### Author Rebuttal · Authors · 2023-08-09
>
> ### Response to Reviewer RpoB
>
> Many thanks to the reviewers for the thorough feedback. We will proceed to respond to each of the comments provided.
>
> **Comment 1:** The datasets used in experiments consists of two-dimensional data. Performance should be evaluated with sparse dataset of much higher dimension.
>
> **Response 1:** We acknowledge the importance of evaluating performance on sparser datasets of higher dimensions. Our initial choice of two-dimensional datasets was driven primarily by their availability and prominence in the literature.
>
> In response to your suggestion, we synthesized a 10-dimensional dataset, termed as "D10". Within this dataset, each function for sampling an interpolation task arises from the summation of several 10-dimensional Gaussian functions, chosen randomly.
>
> We compared the interpolation performance of HINT against existing approaches on this dataset, and the results can be observed in the appended table. Notably, the outcomes demonstrate that the interpolation precision of HINT can effectively scale to datasets with much higher dimensions.
>
> |Interpolation approach|Interpolation accuracy (MSE $\times10^{-4}$) on D10|
> |:---:|:---:|
> |CNP|35.623|
> |ANP|12.578|
> |BANP|12.077|
> |TFR-Transformer|7.465|
> |NIERT|5.496|
> |**HINT**|**4.173**|
>
> Due to space constraints, we are unable to provide a detailed description of the D10 dataset here. The comprehensive construction process of D10 and the associated results will be placed in the supplementary materials.
>
> **Comment 2:** It is difficult to interpret why the masked tokens for the given observed points are necessary.
>
> **Response 2:** The primary intention behind using "masked observed points" is to ensure consistency with the masked target points being input. By doing so, we can obtain re-predictions on the observed points $\hat{y_O}^{(l)}$, which is consistent with the predictions on target points $\hat{y_T}^{(l)}$. This consistency stems from the fact that both predictions emerge from the same function: $\hat{f}(x_{\mathrm{masked}}) = \mathrm{Block}^{(l)}(x_O, y_O, x_{\mathrm{masked}})$ evaluated at both observed and target points.
>
> Consequently, the residuals from re-predictions on the observed points and the residuals from predictions on the target points are strongly correlated. By easily obtaining the former, we can effectively predict the latter.
>
> We appreciate the reviewer's keen observation, and we hope this clarification elucidates our methodology more comprehensively.
>
> **Comment 3:** The local constraint K may be problematic, as it can be dependent on the data properties of each application.
>
> **Response 3:** Indeed, the hyperparameter $K$ for the local constraint is application-dependent, an aspect we have given careful consideration. However, we don't perceive this as "problematic". In our research, the value of $K$ for each dataset was meticulously determined through a series of experiments to ensure optimal performance. Moreover, we validated the efficacy of this parameter through ablation studies. We believe that, with appropriate selection, the constraint remains highly effective. We hope this addresses your concerns.

---

> > ### Comment · Reviewer_RpoB · 2023-08-17
> >
> > Several parts were well addressed in the rebuttal, and I have no more concerns in this work. I'd like to keep the initial rating (borderline accept).

---

> > > ### Author Response · Authors · 2023-08-21
> > > **Response to Reviewer RpoB**
> > >
> > > We appreciate the reviewer's efforts and response. It's gratifying to know that we've successfully addressed and resolved all the concerns raised.

---

### Official Review · Reviewer_G7AZ · 2023-07-07

**Soundness:** 3 good
**Presentation:** 3 good
**Contribution:** 2 fair
**Rating:** 5
**Confidence:** 4

**Summary:**

In this paper, a novel hierarchical residual refining framework called HINT (Hierarchical Residual Refining for Scattered Point Interpolation) was proposed to improve interpolation accuracy. The framework utilized residual information from observed points to guide the prediction of target points in a coarse-to-fine manner, employing correlation modeling. HINT consisted of a series of neural network interpolation blocks, including a main block and multiple residual blocks, which estimated different components of the target function at various scales. Local constraints were applied to the interpolation blocks using K-nearest neighbor graph constraints, conforming to hierarchical residual scales. The hierarchical structure of the main and residual components was leveraged, enabling progressively stronger local constraints to enhance interpolation robustness. Experimental evaluations on diverse theoretical and application scenarios demonstrated the superiority of HINT over existing state-of-the-art interpolation methods in terms of accuracy across various tasks.

**Strengths:**

The paper is well-written and it provides concise descriptions of the proposed framework.

The experiments conducted in this paper are comprehensive.

**Weaknesses:**

A couple of assumptions are used, for example in Sec. 2.3, the authors failed to demonstrate that they all holds in experiments.

The proposed framework is simply a combination of existing modules.

**Questions:**

More literature reviews should be provide to demonstrate the necessity of using Transformer in interpolation.

What are the main differences of the proposed method when compared with  the Transformer encoder-only method in [4]?

The MSE results in Fig.5 in Supplementary Material can be separated into a table for better clarity, and the authors can provide a case by case analysis for those figures in Supplementary Material.

---

> ### Author Rebuttal · Authors · 2023-08-09
>
> ### Response to Reviewer G7AZ
>
> We appreciate the reviewer's constructive feedback. In the following sections, we address each point raised.
>
> **Comment 1:** The assumptions made in Section 2.3 are not validated experimentally.
>
> **Response 1:** In relation to the assumptions highlighted in Section 2.3, we'd like to offer the following clarifications:
>
> 1. **Foundational Work and Validation**:
>    - Our approach is built in the NIERT interpolator [1]. The effectiveness of this methodology is comprehensively discussed in [1], establishing it as a SOTA model for scattered data interpolation.
>
> 2. **Our Innovative Design and Verification**:
>    - **Incorporation of masked observed points**. It has been empirically validated in our ablation study (refer to Table 5, Page 8, third row). Our findings indicated a notable decline in precision when leveraging observed points without the inclusion of "masked observed" points.
>    - **Local constraints on correlation modeling**. This design have been extensively corroborated in Section 4. For instance, Fig. 4 (on Page 8) elucidates how the interpolation blocks in HINT sequentially yield the main component of the target function, succeeded by residuals of diminishing magnitude. Furthermore, Table 5 (on Page 8) reinforces this through ablation studies, indicating that the performance of HINT, when devoid of local constraints or with non-hierarchical local constraints, is inferior to our proposed HINT.
>
> We hope that the above elucidations address your concerns. Should there be any oversight or if further clarifications are required, we humbly solicit your guidance.
>
> **Comment 2:** The proposed framework is simply a combination of existing modules.
>
> **Response 2:** We would like to take this opportunity to further elucidate the unique contributions and the value of our research.
>
> 1. **Beyond a Mere Combination**: While we have indeed leveraged the N-Beats from the time series forecasting and NIERT, our work is not just about juxtaposing two techniques. We have undertaken substantial modifications. Specifically:
>    1) We significantly altered the NIERT block to enable it to output re-predictions of observed points that are consistent with target point predictions.
>    2)  Furthermore, we incorporated local constraints based on KNN Graph, which facilitates adaptation to the varying local scales inherent in function interpolation.
>    3) For the interpolation blocks, we incorporated progressively stringent local constraints. Early blocks focus on capturing the primary components of the function, while subsequent blocks hone in on the intricate details. This hierarchical fashion allows for a nuanced refining of interpolation.
>
> 2. **Pioneering Application of Dual Residual Architecture for scattered Data Interpolation**: Our study is the first to introduce such residual architecture into the realm of scattered data interpolation.
>
> 3. **Superior Experimental Results**: Our method has demonstrated exemplary performance in experiments, achieving state-of-the-art results.
>
>
> **Comment 3:** More literature reviews should be provide to demonstrate the necessity of using Transformer in interpolation.
>
> **Response 3:** Indeed, the existing literature specifically addressing the application of Transformers to scattered data interpolation is somewhat limited. This is the very reason our manuscript heavily references works like NIERT [1] and TFR-Transformer [2]. We can elucidate the superiority of Transformer from the following perspectives:
>
> 1) **Inductive Bias**: The Transformer architecture is particularly suited for handling scattered point data due to its inherent permutation invariance.
>
> 2) **Recent Advancements**: NIERT stands out as one of the most recent and high-performing scattered data interpolation models, effectively showcasing the superiority of Transformer-based models for such tasks.
>
> 3) **Theoretical Underpinnings**: Both references [1] and [3] draw connections between the Transformer framework and traditional interpolation algorithms. Specifically, [1] posits that the self-attention in Transformers can be viewed as a neural representation for interpolation basis function learning. Such perspectives provide a theoretical foundation, underscoring the strong alignment between Transformers and interpolation.
>
> We will endeavor to expand upon and clarify these points in the revised manuscript.
>
>
> **Comment 4:** main differences of the proposed method when compared with the Transformer encoder-only method in [4]?
>
> **Response 4:** NIERT and our proposed HINT exhibit significant differences, which can be summarized as follows:
>
> 1. **Architectural Differences**: The Transformer encoder-only method, termed as NIERT, employs a singular Transformer encoder as its main model. In contrast, our proposed methodology, named HINT, integrates a series of lightweight blocks, rooted in the Transformer encoder architecture, and connected through residual links.
>
> 2. **Methodological Variations**: NIERT directly predict target point values using a whole piece model. Our HINT adopts a hierarchical refining process. It firstly predicts the main component of the function, followed by a series of residual components incrementally.
>
> **Comment 5:** The MSE results in Fig.5 in Supplementary Material can be separated into a table for better clarity...
>
> **Response 5:** For clarity, we will adjust the presentation in accordance with the reviewer's suggestions.
>
> ---
>
> [1] Shizhe Ding and Dongbo Bu. NIERT: Accurate Numerical Interpolation through Unifying Scattered Data Representations using Transformer Encoder. arXiv preprint arXiv:2209.09078, 2023.
>
> [2] Xiaoqian Chen, Zhiqiang Gong, Xiaoyu Zhao, Weien Zhou, and Wen Yao. A Machine Learning Modelling Benchmark for Temperature Field Reconstruction of Heat-Source Systems. arXiv preprint arXiv:2108.08298, 2021.
>
> [3] Cao, Shuhao, Peng Xu, and David A. Clifton. How to understand masked autoencoders. arXiv preprint arXiv:2202.03670, 2022.

---

> > ### Comment · Reviewer_G7AZ · 2023-08-17
> >
> > Based on the rebuttal made in response to my comments, the authors have adequately addressed the majority of the concerns raised.

---

> > > ### Author Response · Authors · 2023-08-21
> > > **Response to Reviewer G7AZ**
> > >
> > > We sincerely thank the reviewer for the efforts and feedback. We are heartened to hear that we have adequately addressed the majority of the concerns raised.

---

### Official Review · Reviewer_746Q · 2023-07-22

**Soundness:** 2 fair
**Presentation:** 3 good
**Contribution:** 3 good
**Rating:** 5
**Confidence:** 3

**Summary:**

The paper introduces the Hierarchical INTerpolation Network (HINT) as an accurate interpolation algorithm for various theoretical and engineering applications. HINT consists of lightweight interpolation blocks arranged sequentially, where the first block estimates the main component of the target function, and subsequent blocks predict residual components using observed point residuals from preceding blocks.  Moreover, the authors introduce hierarchical local constraints. Extensive experiments demonstrate that HINT significantly outperforms existing interpolation algorithms.

**Strengths:**

1. The method that utilizing the first and subsequent blocks to estimate the main component and residual components is novel. It applies the residual method to  interpolation algorithm, which is also interesting.
2. The whole method is easy to follow.
3. This paper conduct extensive extriments on various datasets, demonstrateing its superiority.

**Weaknesses:**

1. The comparison of the time cost is missing. It's necessary to compare the proposed HINT and other methods including the traditional ones and deep learning ones.
2. The paper lacks the ablation study of hyper-parameter L and loss scale $\lambda$. Besides, does the value of $\lambda$ keep the same for all the datasets?
3. For Mathit-2D and Perlin, the choich of L and K is (2,6) and (4,2), respectively. Could the author provide some guidlines for the choice ? Or it may Hinder the spread and use of this method.

**Questions:**

see above

**Limitations:**

yes

---

> ### Author Rebuttal · Authors · 2023-08-09
>
> ### Response to Reviewer 746Q
>
> We sincerely thank Reviewer 746Q for the insightful comments and suggestions. We will address each comment in the following.
>
> **Comment 1:** The absence of time cost comparison between HINT and other methods and comparison with traditional methods.
>
> **Response 1:**
> In response to the comment regarding the comparison of time cost, we have performed additional evaluations. Specifically, we assessed the average interpolation time cost of our proposed HINT method and existing interpolation techniques, both traditional and deep-learning-based, on the Mathit-2D test set. Following the work from [1], we selected RBF [2] and MIR [3] as representatives of traditional algorithms. The results are tabulated below:
>
> |Interpolation approach| Average time (ms)|Interpolation accuracy (MSE $\times10^{-4}$)|
> |:---:|:---:|:---:|
> |RBF|0.66|34.706|
> |MIR|80.30|27.460|
> |CNP|13.53|24.868|
> |ANP|27.55|14.001|
> |BANP|29.19|8.419|
> |TFR-Transformer|26.16|5.857|
> |NIERT|31.99|3.167|
> |**HINT**|36.20|**2.722**|
>
> As evident from the table, the traditional interpolation method, RBF, offers the highest interpolation efficiency at 0.66ms, but its accuracy is substantially compromised. MIR offers slightly better accuracy, it possesses the longest interpolation time amongst all methods evaluated (80.30ms). For deep neural network-based methods, there is a clear trend of increasing accuracy from CNP to our HINT method and the time cost also gradually rises. Specifically, our HINT method has an average time of 36.20 ms, which, while marginally higher than that of NIERT (31.99ms), is considerably more efficient than the traditional MIR (80.30ms).
>
> Moreover, it's worth noting that, our current HINT implementation employs dense attention computation combined with an attention mask to realize the hierarchical local constraints proposed in our paper. Given that these local constraints are theoretically sparse, there remains room for optimizing the time cost of HINT. Our primary focus in this research was to achieve high interpolation accuracy, and, as of now, HINT appears to be best suited for scenarios demanding high accuracy while being somewhat lenient on time constraints.
>
> **Comment 2:** lacks of the ablation study of hyper-parameter $L$ and loss scale $\lambda$.
>
> **Response 2:** In fact, we conducted ablation studies on the hyper-parameter $L$ and loss scale $\lambda$. The ablation study concerning the block number $L$ can be found in Table 6 on Page 8. The ablation experiment for the auxiliary loss weight $\lambda$ is presented in the first row of Table 5 on Page 8, specifically under the section "HINT w/o $\mathcal{L}_{\mathrm{aux}}$", where this implies $\lambda=0$. To clarify further on your query regarding the value of $\lambda$, it does vary, and these specific values are detailed in the last row of Table 2 on Page 3 in the supplementary material. We arrived at these values after a series of experiments to fine-tune our approach.
>
> We appreciate your question as it has made us realize that we might not have explicitly highlighted these details in our current presentation. We will emphasize these more prominently in the revised version of our manuscript to prevent any ambiguity.
>
> **Comment 3:** The guidelines for choosing hyperparameters (block number $L$, attention layer number, and local parameter $K$).
>
> **Response 3:** If I understand correctly, the second hyperparameter you mentioned is the number of attention layers in the main block. In fact, the choice of these hyperparameters is not arbitrary. We outline the rationale for setting these parameters as follows:
>
> 1. **Block number ($L$)**: In our experiments, we observed that for functions with smoother profiles and fewer details within a set, a smaller block number $L$ is appropriate. Conversely, for functions rich in details and containing higher frequency components, a larger block number $L$ is more suitable.
>
> 2. **Attention layers in the main block**: Concerning the number of attention layers in the main block, our findings indicate that larger datasets demand more attention layers to attain sufficient model capacity. On the other hand, for smaller datasets, fewer attention layers can help prevent overfitting.
>
> 3. **Local constraint parameter ($K$)**: Setting $K$ is somewhat challenging, requiring insights into the data or even expert knowledge. Such knowledge encompasses spatial auto-correlation of the function. If the function exhibits auto-correlation over a larger neighborhood, a larger $K$ might be more appropriate. In our experiments, we found that determining the optimal $K$ often requires trial and fine-tuning.
>
> For instance, for the Mathit dataset—a synthetic set comprised of smoother mathematical functions with a significant volume—a smaller block number $L$ and more attention layers seem optimal. The mathematical functions' features correlate over longer distances (periodicity, symmetry, trends, etc.), thus initially a larger $K$ covering all observation points is prudent. Conversely, for the Perlin dataset—a real-world 2D velocity field dataset of smaller volume but intricate, detailed functions—a larger block number $L$, fewer attention layers, and a smaller $K$ are logical. We appreciate the reviewer's constructive feedback and will incorporate this discussion in the Supplementary Material.
>
> ------
>
> [1] Shizhe Ding and Dongbo Bu. NIERT: Accurate Numerical Interpolation through Unifying Scattered Data Representations using Transformer Encoder. arXiv preprint arXiv:2209.09078, 2023.
>
> [2] M. J. Powell, Radial basis functions for multivariable interpolation: a review. Algorithms for Approximation, 1987.
>
> [3] Q. Wang, P. Moin, and G. Iaccarino, A high order multivariate approximation scheme for scattered data sets, Journal of Computational Physics, vol. 229, no. 18, pp. 6343–6361, 2010.

---

> > ### Comment · Reviewer_746Q · 2023-08-13
> > **Response to Rebuttal.**
> >
> > We thank the author for rebuttals.
> >
> > The comparison of the time cost seems fine. My main concerns are addressed.

---

> > > ### Author Response · Authors · 2023-08-21
> > > **Response to Reviewer 746Q**
> > >
> > > We are grateful for the reviewer's understanding and acknowledgment of our rebuttals. It's reassuring to know that we have successfully addressed your main concerns. We thank the reviewer for the constructive feedback throughout this review process.

---

### Author Rebuttal · Authors · 2023-08-09

### Supplementary Result to Reviewer WnAW

We evaluated the impact of varying observed point counts on interpolation accuracy using the Mathit-2D dataset, as shown in the supplementary figure. On Mathit-2D, both our HINT method and other techniques exhibit a marked reduction in interpolation error (MSE) as the number of observed points increases. Notably, HINT outperforms other methods across all observed point settings, underscoring its superior performance.

---

### Decision · Program_Chairs · 2023-09-21

**Decision:**

Accept (poster)

**Comment:**

The paper introduces HINT, a novel hierarchical residual refining framework for interpolation tasks. Reviewers appreciate its unique aspects, including the use of initial and subsequent blocks for estimation and the application of residual methods to interpolation. The paper's clarity and comprehensive experimental evaluations are strengths. However, concerns about novelty, method simplicity, lack of comparisons, and experimental limitations were raised. After the rebuttal, additional comparisons and clarifications have effectively addressed these concerns. The reviewers now recommend acceptance, given the method's effectiveness while maintaining efficiency, demonstrating practical potential. We have decided to accept this submission after thorough discussion. We encourage the author to thoughtfully incorporate reviewers' feedback into the final version, particularly regarding the clarification of related works.